# Learning Soft Constraints From Constrained Expert Demonstrations

**Ashish Gaurav[1,2], Kasra Rezaee[3], Guiliang Liu[4], Pascal Poupart[1,2]**
a5gaurav@uwaterloo.ca, kasra.rezaee@huawei.ca,
liuguiliang@cuhk.edu.cn, ppoupart@uwaterloo.ca

[1]Cheriton School of Computer Science, University of Waterloo, Canada
[2]Vector Institute, Toronto, Canada
[3]Huawei Technologies Canada
[4]School of Data Science, The Chinese University of Hong Kong, Shenzhen

## Abstract

Inverse reinforcement learning (IRL) methods assume that the expert data is generated by an agent optimizing some reward function. However, in many settings, the agent may optimize a reward function subject to some constraints, where the constraints induce behaviors that may be otherwise difficult to express with just a reward function. We consider the setting where the reward function is given, and the constraints are unknown, and propose a method that is able to recover these constraints satisfactorily from the expert data. While previous work has focused on recovering hard constraints, our method can recover cumulative soft constraints that the agent satisfies on average per episode. In IRL fashion, our method solves this problem by adjusting the constraint function iteratively through a constrained optimization procedure, until the agent behavior matches the expert behavior. We demonstrate our approach on synthetic environments, robotics environments and real world highway driving scenarios.

## 1 Introduction

Inverse reinforcement learning (IRL) (Ng et al., 2000; Russell, 1998) refers to the problem of learning a reward function given observed optimal or near optimal behavior. However, in many setups, expert actions may result from a policy that inherently optimizes a reward function subject to certain constraints. While IRL methods are able to learn a reward function that explains the expert demonstrations well, many tasks also require knowing constraints. Constraints can often provide a more interpretable representation of behavior than just the reward function (Chou et al., 2018). In fact, constraints can represent safety requirements more strictly than reward functions, and therefore are especially useful in safety-critical applications (Chou et al., 2021; Scobee & Sastry, 2019).

**Inverse constraint learning (ICL)** may therefore be defined as the process of extracting the constraint function(s) associated with the given optimal (or near optimal) expert data, where we assume that the reward function is available. Notably, some prior work (Chou et al., 2018; 2020; 2021; Scobee & Sastry, 2019; Malik et al., 2021) has tackled this problem by learning hard constraints (i.e., functions that indicate which state action-pairs are allowed).

We propose a novel method for ICL (for simplicity, our method is also called ICL) that learns cumulative *soft constraints* from expert demonstrations while assuming that the reward function is known. The difference between hard constraints and soft constraints can be illustrated as follows. Suppose in an environment, we need to obey the constraint "do not use more than 3 units of energy". As a hard constraint, we typically wish to ensure that this constraint is always satisfied for any individual trajectory. The difference between this "hard" constraint and proposed "soft" constraints is that soft constraints are not necessarily satisfied in every trajectory, but rather only satisfied in expectation. This is equivalent to the specification "on average across all trajectories, do not use more than 3 units of energy". In the case of soft constraints, there may be certain trajectories when the constraint is violated, but in expectation, it is satisfied.

To formulate our method, we adopt the framework of constrained Markov decision processes (CMDP) (Altman, 1999), where an agent seeks to maximize expected cumulative rewards subject to constraints on the expected cumulative value of constraint functions. While previous work in constrained RL focuses on finding an optimal policy that respects known constraints, we seek to learn the constraints based on expert demonstrations. We adopt an approach similar to IRL, but the goal is to learn the constraint functions instead of the reward function.

**Contributions.** Our contributions can be summarized as follows: (a) We propose a novel formulation and method for ICL. Our approach works with any state-action spaces (including continuous state-action spaces) and can learn arbitrary constraint functions represented by flexible neural networks. To the best of our knowledge, our method is the first to learn *cumulative soft constraints* (such constraints can take into account noise in sensor measurements and possible violations in expert demonstrations) bounded in expectation as in constrained MDPs. (b) We demonstrate our approach by learning constraint functions in various synthetic environments, robotics environments and real world highway driving scenarios.

The paper is structured as follows. Section 2 provides some background about IRL and ICL. Section 3 summarizes previous work about constraint learning. Section 4 describes our new technique to learn cumulative soft constraints from expert demonstrations. Section 5 demonstrates the approach for synthetic environments and discusses the results (*more results are provided in Appendix B*). Finally, Section 6 concludes by discussing limitations and future work.

## 2    BACKGROUND

**Markov Decision Process (MDP).** An MDP is defined as a tuple $(\mathcal{S}, \mathcal{A}, p, \mu, r, \gamma)$, where $\mathcal{S}$ is the state space, $\mathcal{A}$ is the action space, $p(\cdot|s, a)$ are the transition probabilities over the next states given the current state $s$ and current action $a$, $r : \mathcal{S} \times \mathcal{A} \to \mathbb{R}$ is the reward function, $\mu : \mathcal{S} \to [0, 1]$ is the initial state distribution and $\gamma$ is the discount factor. The behavior of an agent in this MDP can be represented by a stochastic policy $\pi : \mathcal{S} \times \mathcal{A} \to [0, 1]$, which is a mapping from a state to a probability distribution over actions. A *constrained MDP* augments the MDP structure to contain a constraint function $c : \mathcal{S} \times \mathcal{A} \to \mathbb{R}$ and an episodic constraint threshold $\beta$.

**Reinforcement learning and Constrained RL.** The objective of any standard RL procedure (control) is to obtain a policy that maximizes the (infinite horizon) expected long term discounted reward (Sutton & Barto, 2018):

$$\pi^* = \arg\max_{\pi} \mathbb{E}_{s_0 \sim \mu(\cdot), a_t \sim \pi(\cdot|s_t), s_{t+1} \sim p(\cdot|s_t, a_t)} \left[ \sum_{t=0}^{\infty} \gamma^t r(s_t, a_t) \right] =: J_{\mu}^{\pi}(r) \tag{1}$$

Similarly, in constrained RL, additionally the expectation of cumulative constraint functions $c_i$ must not exceed associated thresholds $\beta_i$:

$$\pi^* = \arg\max_{\pi} J_{\mu}^{\pi}(r) \text{ such that } J_{\mu}^{\pi}(c_i) \leq \beta_i \forall i \tag{2}$$

For simplicity, in this work we consider constrained RL with only one constraint function.

**Inverse reinforcement learning (IRL) and inverse constraint learning (ICL).** IRL performs the inverse operation of reinforcement learning, that is, given access to a dataset $\mathcal{D} = \{\tau_j\}_{j=1}^N = \{\{(s_t, a_t)\}_{t=1}^{M_j}\}_{j=1}^N$ sampled using an optimal or near optimal policy $\pi^*$, the goal is to obtain a reward function $r$ that best explains the dataset. By "best explanation", we mean that if we perform the RL procedure using $r$, then the obtained policy captures the behavior demonstrated in $\mathcal{D}$ as closely as possible. In the same way, given access to a dataset $\mathcal{D}$ (just like in IRL), which is sampled using an optimal or near optimal policy $\pi^*$ (respecting some constraints $c_i$ and maximizing some known reward $r$), the goal of ICL is to obtain the constraint functions $c_i$ that best explain the dataset, that is, if we perform the constrained RL procedure using $r, c_i \forall i$, then the obtained policy captures the behaviour demonstrated in $\mathcal{D}$.

**Setup.** Similar to prior work (Chou et al., 2020), we learn only the constraints, but not the reward function. Essentially, it is difficult to say whether a demonstrated behaviour is obeying a constraint, or maximizing a reward, or doing both. So, for simplicity, we assume the (nominal) reward is given, and

we just need to learn a (single) constraint function. Without loss of generality, we fix the threshold $\beta$ to a predetermined value and learn only a constraint function $c$.[1]

# 3 RELATED WORK

Several works in the inverse RL literature consider the framework of constrained MDPs (Kalweit et al., 2020; Fischer et al., 2021; Ding & Xue, 2022). However, those works focus on learning the reward function while assuming that the constraints are known. In contrast, we focus on learning constraints while assuming that the reward function is known.

Initial work in constraint learning focused on learning instantaneous constraints with respect to states and actions. When states and actions are discrete, constraint sets can be learned to distinguish feasible state-action pairs from infeasible ones (Chou et al., 2018; Scobee & Sastry, 2019; McPherson et al., 2021; Park et al., 2020). In continuous domains, various types of constraint functions are learned to infer the boundaries of infeasible state-action pairs. This includes geometric state-space constraints (Armesto et al., 2017; Pérez-D'Arpino & Shah, 2017), task space equality constraints (Lin et al., 2015; 2017), convex constraints (Menner et al., 2021) and parametric non-convex constraint functions (Chou et al., 2020; Malik et al., 2021). Several other works learn local trajectory-based constraints with respect to a single trajectory (Calinon & Billard, 2008; Pais et al., 2013; Li & Berenson, 2016; Mehr et al., 2016; Ye & Alterovitz, 2017).

The vast majority of previous work considers hard constraints, which is fine in deterministic domains. However, in stochastic domains, the probability of violating a constraint can rarely be reduced to zero, making hard constraints inadequate. To that effect, Glazier et al. (2021) learn probabilistic constraints that hold with high probability in expert demonstrations. This approach extends the framework of maximum entropy inverse reinforcement learning to learn a constraint value that is treated as a negative reward added to the given reward function. Since the probability of a trajectory is proportional to the exponential of the rewards, this has the effect of reducing the probability of trajectories with high constraint values. Several other works also extend the inverse entropy RL framework to learn constraints that reduce some transition probabilities to 0 when a constraint is violated, but this amounts to hard constraints again (Scobee & Sastry, 2019; Park et al., 2020; McPherson et al., 2021; Malik et al., 2021). In another line of work, Bayesian approaches (Chou et al., 2021; Papadimitriou et al., 2021) have also been proposed to learn a distribution over constraints due to the unidentifiability and uncertainty of the true underlying constraints. With the exception of (Papadimitriou et al., 2021), existing techniques that learn probabilistic constraints are restricted to discrete states and actions. Furthermore, the probabilistic constraints that are learned do not correspond to the soft constraints of constrained MDPs. Hence, we fill this gap by proposing a first technique that learns soft cumulative constraint functions bounded in expectations (as in constrained MDPs) for stochastic domains with any state-action spaces.

# 4 APPROACH

## 4.1 INVERSE CONSTRAINT LEARNING: TWO PHASES

We propose an approach to do ICL, that is, given a reward $r$ and demonstrations $\mathcal{D}$, this strategy obtains a constraint function $c$ such that when $r, c$ are used in the constrained RL procedure (any algorithm that optimizes Equation (2)), the obtained policy $\pi^*$ explains the behavior in $\mathcal{D}$. Our approach is based on the template of IRL, where we typically alternate between a policy optimization phase and a reward adjustment phase (Arora & Doshi, 2021). We first describe a theoretical procedure that captures the essence of our approach and then later adapt it into a practical algorithm. The theoretical approach starts with an empty set of policies (i.e., $\Pi = \emptyset$) and then grows this set of

---

[1]Mathematically equivalent constraints can be obtained by multiplying the constraint function and the threshold by the same value. Therefore there is no loss in fixing $\beta$ to learn a canonical constraint within the set of equivalent constraints, assuming that the learned constraint function can take arbitrary values.

policies by alternating between two optimization procedures until convergence:

$$\text{POLICY OPTIMIZATION: } \pi^* := \arg\max_\pi J_\mu^\pi(r) \text{ such that } J_\mu^\pi(c) \leq \beta \text{ and } \Pi \leftarrow \Pi \cup \{\pi^*\} \quad (3)$$

$$\text{CONSTRAINT ADJUSTMENT: } c^* := \arg\max_c \min_{\pi \in \Pi} J_\mu^\pi(c) \text{ such that } J_\mu^{\pi_E}(c) \leq \beta \quad (4)$$

The optimization procedure in Equation (3) performs forward constrained RL (see Section 2 - Background and Equation (2) for the definition of constrained RL and the notation $J_\mu^\pi$) to find an optimal policy $\pi^*$ given a reward function $r$ and a constraint function $c$. This optimal policy is then added to the set of policies $\Pi$. This is followed by the optimization procedure in Equation (4), which adjusts the constraint function $c$ to increase the constraint values of the policies in $\Pi$ while keeping the constraint value of the expert policy $\pi_E$ bounded by $\beta$. This is achieved by maximizing the accumulated constraint value for the most feasible (lowest total constraint value) policy in $\Pi$. This selectively increases the constraint function until the most feasible optimal policies become infeasible. At each iteration of those two optimization procedures a new policy is found but its constraint value will be increased past $\beta$ unless it corresponds to the expert policy (which is enforced by the constraint in Equation (4)). By doing so, this approach will converge to the expert policy (or an equivalent policy when multiple policies can generate the same trajectories) as shown in Theorem 1. Intuitively, this happens because all policies and trajectories except the expert's are made infeasible in the long run.

**Theorem 1.** *Assuming there is a unique policy $\pi^E$ that achieves $J_\mu^{\pi^E}(r)$, the alternation of optimization procedures in Equation (3) and Equation (4) converges to a set of policies $\Pi$ such that the last policy $\pi^*$ added to $\Pi$ is equivalent to the expert policy $\pi^E$ in the sense that $\pi^*$ and $\pi^E$ generate the same trajectories.* (For proof, see Appendix A)

In practice, there are several challenges in implementing the optimization procedures proposed in Equation (3) and Equation (4). First, we do not have the expert policy $\pi_E$, but rather trajectories generated based on the expert policy. Also, the set $\Pi$ of policies can grow to become very large before convergence is achieved. Furthermore, convergence may not occur or may occur prematurely due to numerical issues and whether the policy space contains the expert policy. The optimization procedures include constraints and one of them requires min-max optimization. Therefore, we approximate the theoretical approach in Equation (3) and Equation (4) into the practical approach described in Algorithm 1.

As a first step, we replace the optimization procedure in Equation (4) by a simpler optimization described in Equation (5). More precisely, we replace the max-min optimization (which is a continuous-discrete optimization problem) of the constraint values of the policies in $\Pi$ by a maximization of the constraint value of the mixture $\pi_{mix}$ of policies in $\Pi$. In this case, the mixture $\pi_{mix}$ is a collection of optimal policies where each policy has a weight, which is used in the computation of $J_\mu^{\pi_{mix}}(c)$ in Equation (5). The details of this computation are stated in Algorithm 3. Overall, changing the objective avoids a challenging max-min optimization, but we lose the theoretical guarantee of convergence to a policy equivalent to the expert policy. However, we find in our experiments that the algorithm still converges empirically.

$$c^* := \arg\max J_\mu^{\pi_{mix}}(c) \text{ such that } J_\mu^{\pi_E}(c) \leq \beta \quad (5)$$

Maximizing the constraint values of a mixture of policies $\Pi$ usually tends to increase the constraint values for all policies in $\Pi$ most of the time, and when a policy's constraint value is not increased beyond $\beta$ it will usually be a policy close to the expert policy.

## 4.2 SOLVING THE CONSTRAINED OPTIMIZATIONS THROUGH THE PENALTY METHOD

The constrained optimization problems Equation (3), Equation (4), Equation (5), described in the preceding subsection belong to the following general class of optimization problems (here, $f, g$ are potentially non-linear and non-convex):

$$\min_y f(y) \text{ such that } g(y) \leq 0 \quad (6)$$

While many existing constrained RL algorithms formulate constrained optimization from a Lagrangian perspective (Borkar, 2005; Bhatnagar & Lakshmanan, 2012; Tessler et al., 2018; Bohez et al., 2019) as a min-max problem that can be handled by gradient ascent-descent type algorithms, Lagrangian

formulations are still challenging in terms of empirical convergence, notably suffering from oscillatory behaviour (demonstrated for Equation (5) in Appendix B.4).

Therefore, we use a less commonly used optimization framework, namely, the **penalty method** (Bertsekas, 2014; Donti et al., 2020) which converts a constrained problem into an unconstrained problem with a non differentiable RELU term. Roughly, the approach starts by instantiating $y = y_0$, and then two (sub) procedures are repeated for a few steps each until convergence: (a) first a *feasible* solution is found by repeatedly modifying $y$ in the direction of $-\nabla g(y)$ until $g(y) \leq 0$ (note that this is equivalent to $-\nabla ReLU(g(y))$), also called *feasibility projection*, then (b) a soft loss is optimized that simultaneously minimizes $f(y)$ while trying to keep $y$ within the feasible region. This soft loss is as follows: (here, $\lambda$ is a hyperparameter):

$$\min_y L_{\text{soft}}(y) := f(y) + \lambda \text{ReLU}(g(y)) \tag{7}$$

As an alternative to Lagrangian based approaches, the penalty method has the advantage of simpler algorithmic implementation. Further, as we will demonstrate later in our experiments, it also performs well empirically.

Note that the choice of $\lambda$ is crucial here. In fact, we plan to use the penalty method to optimize both Equation (3) and Equation (5), using different $\lambda$ for each case. A small $\lambda$ means that in the soft loss optimization subprocedure (Equation (7)), the gradient update of $\text{ReLU}(g(y))$ is miniscule in comparison to the gradient update of $f(y)$, because of which $y$ may not stay within the feasible region during the soft loss optimization. In this case, the feasibility projection subprocedure is important to ensure the overall requirement of feasibility is met (even if exact feasibility cannot be guaranteed). Conversely, a large $\lambda$ means that minimizing the soft loss is likely to ensure feasibility, and the feasibility projection subprocedure may therefore be omitted. This forms the basis for our proposed optimization procedures (see Algorithms 2 and 3 for the exact steps of these procedures). Namely, we perform forward constrained RL (Equation (3)) using $\lambda = 0$ with the feasibility projection subprocedure, and constraint adjustment (Equation (4) / Equation (5)) using a moderate/high $\lambda$ and without the feasibility projection subprocedure.

It should be noted that out of the two procedures represented by Equation (3) and Equation (5), the constrained RL procedure can still be replaced by any equivalent algorithm that solves Equation (3) efficiently. Even in that case, the primary optimization objectives remain the same as expressed in Equation (3), Equation (4) and Equation (5). However, our preferred method for solving Equation (5) is using a penalty formulation as described.

### 4.3 IMPROVING CONSTRAINT ADJUSTMENT

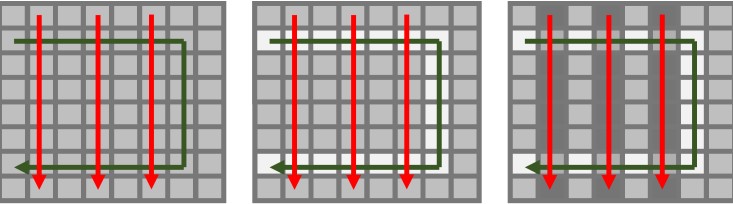

Figure 1: Given expert (green) and agent (red) behaviour for an arbitrary Gridworld environment (first figure), the proposed algorithm decreases the constraint value for the expert behaviour (second figure) and increases the constraint value for the agent behaviour (third figure). Constraint learning is slower if there is overlap between the two behaviours since there exist some states for which the algorithm tries to simultaneously increase and decrease the constraint value. Note that dark gray represents a higher constraint value, light gray a medium constraint value and white represents a lower constraint value.

We can further improve the constraint adjustment subprocedure by noticing that an overlap in expert and agent data used in computing the terms $J_\mu^{\pi_E}(c)$ and $J_\mu^{\pi_{mix}}(c)$ respectively in Equation (5) hampers convergence in learning constraints. We can see this in the following way. We know that constraint adjustment (Equation (4), Equation (5)) is finding the decision boundary between expert and agent trajectories (see Figure 1 for an illustration). Consider the soft loss objective for Equation (5):

$$\min_c L_{\text{soft}}(c) := -J_\mu^{\pi_{mix}}(c) + \lambda \text{ReLU}(J_\mu^{\pi_E}(c) - \beta)$$

Here, the expert data is used to compute $J_\mu^{\pi_E}(c)$, and the agent data is used to compute $J_\mu^{\pi_{mix}}(c)$. Depending on whether $J_\mu^{\pi_E}(c) - \beta \leq 0$ or not, the ReLU term vanishes in $L_{\text{soft}}(c)$. There are two cases. **(I)** If $J_\mu^{\pi_E}(c) - \beta \leq 0$, then $c$ is already feasible, that is, for this value of $c$, the average constraint value across expert trajectories is less than or equal to $\beta$. If there are expert (or expert-like) trajectories in agent data (used to compute $-J_\mu^{\pi_{mix}}(c)$), then we will then end up increasing the constraint value across these expert trajectories (on average), which is not desirable since it will lead to $c$ becoming more infeasible. This will lead to requiring more iterations to converge in learning the constraint function $c$. **(II)** If $J_\mu^{\pi_E}(c) - \beta > 0$, then there is a nonzero ReLU term in $L_{\text{soft}}(c)$. If there are some expert or expert-like trajectories in agent data (and subsequently in $-J_\mu^{\pi_{mix}}(c)$), and we take the gradient of $L_{\text{soft}}(c)$, we will get two contrasting gradient terms trying to increase and decrease the constraint value across the same expert (or expert-like) trajectories. The gradient update associated with the ReLU term is required since we want $c$ to become feasible, but having expert (or expert-like) trajectories in $-J_\mu^{\pi_{mix}}(c)$ diminishes the effect of the ReLU term and then more iterations are needed for convergence of constraint function $c$.

To improve empirical convergence, we propose two reweightings: (a) we reweight the policies in $\pi_{mix}$ (line 6 in Algorithm 3) so that policies dissimilar to the expert policy are favoured more in the calculation of $-J_\mu^{\pi_{mix}}(c)$, and (b) we reweight the individual trajectories in the expectation $-J_\mu^{\pi_{mix}}(c)$ (line 8 in Algorithm 3) to ensure that there is less or negligible weight associated with the expert or expert-like trajectories. We can perform both these reweightings using a density estimator (specifically, RealNVP flow (Dinh et al., 2016)). The idea is to learn the density of expert or expert-like state-action pairs and compute the negative log-probability (NLP) of any given trajectory's state-action pairs at test time to determine if it is expert or expert-like, or not. Practically, this is estimated by computing the mean and std. deviation of the NLP of expert state-action pairs, and then at test time, checking if the NLP of the given state-action pairs is within one std. deviation of the mean or not. Please refer to lines 1-3 in Algorithm 3 for the exact pseudocode.

Our complete algorithm is provided in Algorithm 1. As mentioned in Section 4.1, the algorithm repeatedly optimizes the objectives defined in Equation (3) and Equation (5) on lines 6 and 7. The procedures for these optimizations are provided in Algorithms 2 and 3, which use the penalty method as described in Section 4.2. Further, Algorithm 3 describes constraint adjustment using a normalizing flow as described in this subsection. *Inputs and outputs to each algorithm have also been specified.*

---

**Algorithm 1** INVERSE-CONSTRAINT-LEARNING

---
    **hyper-parameters:** number of ICL iterations $n$, tolerance $\epsilon$
    **input:** expert dataset $\mathcal{D} = \{\tau\}_{\tau \in \mathcal{D}} := \{\{(s_t, a_t)\}_{1 \leq t \leq |\tau|}\}_{\tau \in \mathcal{D}}$
1:  **initialize** normalizing flow $f$
2:  **optimize** likelihood of $f$ on expert state action data: $\max_f \text{SUM}_{(s,a) \in \tau, \tau \in \mathcal{D}}(\log p_f(s, a))$
3:  **initialize** constraint function $c$ (parameterized by $\phi$)
4:  **for** $1 \leq i \leq n$ **do**
5:      **initialize** policy $\pi_i$ (parameterized by $\boldsymbol{\theta}_i$)
6:      **perform** $\pi_i := $ CONSTRAINED-RL$(\pi_i, c)$
7:      **perform** $c := $ CONSTRAINT-ADJUSTMENT$(\pi_{1:i}, c, \mathcal{D}, f)$
8:      **break** if NORMALIZED-ACCRUAL-DISSIMILARITY$(\mathcal{D}, \mathcal{D}_{\pi_i}) \leq \epsilon$
                                   $\triangleright$ *See Section 5 for normalized accrual dissimilarity metric*
9:  **end for**
    **output:** learned constraint function $c$ (neural network with sigmoid output),
            learned most recent policy $\pi_i$

---

## 5 EXPERIMENTS

**Environments.** We conduct several experiments on the following environments: (a) *Gridworld (A, B)*, which are 7x7 gridworld environments, (b) *CartPole (MR or Move Right, Mid)* which are variants of the CartPole environment from OpenAI Gym (Brockman et al., 2016), (c) *Highway driving environment* based on the HighD dataset (Krajewski et al., 2018), (d) *Mujoco robotics environments (Ant-Constrained, HalfCheetah-Constrained)*, and (e) *Highway lane change environment* based on the ExiD dataset (Moers et al., 2022). Due to space requirements, we provide further details on all these environments (figures and explanation) in Appendix C.1.

---

**Algorithm 2** CONSTRAINED-RL

---

    **hyper-parameters:** learning rates $\eta_1, \eta_2$, constraint threshold $\beta$, constrained RL epochs $m$
    **input:** policy $\pi_i$ parameterized by $\boldsymbol{\theta}_i$, constraint function $c$
1: **for** $1 \leq j \leq m$ **do**
2:     **correct** $\pi_i$ to be feasible: (iterate) $\boldsymbol{\theta}_i \leftarrow \boldsymbol{\theta}_i - \eta_1 \nabla_{\boldsymbol{\theta}_i} \text{RELU}(J_\mu^{\pi_i}(c) - \beta)$
3:     **optimize** expected discounted reward: $\boldsymbol{\theta}_i \leftarrow \boldsymbol{\theta}_i - \eta_2 \nabla_{\boldsymbol{\theta}_i} \text{PPO-LOSS}(\pi_i)$
                                   ▷ *Proximal Policy Optimization* (Schulman et al., 2017)
4: **end for**
    **output:** learned policy $\pi_i$

---

**Algorithm 3** CONSTRAINT-ADJUSTMENT

---

    **hyper-parameters:** learning rate $\eta_3$, penalty wt. $\lambda$, constraint threshold $\beta$,
            constraint adjustment epochs $e$
    **input:** policies $\pi_{1:i}$, constraint function $c$, trained normalizing flow $f$,
            expert dataset $\mathcal{D} = \{\tau\}_{\tau \in \mathcal{D}} := \{\{(s_t, a_t)\}_{1 \leq t \leq |\tau|}\}_{\tau \in \mathcal{D}}$
    **given:** $c^\gamma(\tau) := \text{SUM}_{1 \leq t \leq |\tau|}(\gamma^{t-1} c(s_t, a_t))$,
            $\text{SAMPLE}_\tau(\Pi, p)$ which generates $|\mathcal{D}|$ trajectories $\tau = \{(s_t, a_t)\}_{1 \leq t \leq |\tau|}$, where for each
            $\tau$, we choose $\pi \in \Pi$ with prob. $p(\pi)$, then, $s_1 \sim \mu(\cdot), a_t \sim \pi(\cdot|s_t), s_{t+1} \sim p(\cdot|s_t, a_t)$
1: $\mu_E := \text{MEAN}_{(s,a) \in \tau, \tau \in \mathcal{D}}(-\log p_f(s, a))$
2: $\sigma_E := \text{STD-DEV}_{(s,a) \in \tau, \tau \in \mathcal{D}}(-\log p_f(s, a))$
3: $w(\tau) := \text{MEAN}_{(s,a) \in \tau}(\mathbf{1}(-\log p_f(s, a) > \mu_E + \sigma_E))$    ▷ *trajectory dissimilarity w.r.t. expert*
4: **construct** policy dataset $\mathcal{D}_{\pi_i} = \text{SAMPLE}_\tau(\Pi = \{\pi_i\}, p = \{1\})$
5: $\tilde{w}_i := \text{MEAN}_{\tau \in \mathcal{D}_{\pi_i}} w(\tau)$                       ▷ *unnormalized policy weights*
6: **construct** agent dataset $\mathcal{D}_A = \text{SAMPLE}_\tau(\Pi = \pi_{1:i}, p(\pi_i) \propto \tilde{w}_i)$        ▷ *policy reweighting*
7: **for** $1 \leq j \leq e$ **do**                           ▷ *constraint function adjustment*
8:     **compute** $J_\mu^{\pi_{mix}}(c) := \text{SUM}_{\tau \in \mathcal{D}_A} \frac{w(\tau) c^\gamma(\tau)}{\text{SUM}_{\tau \in \mathcal{D}_A} w(\tau)}$        ▷ *trajectory reweighting*
9:     **compute** $J_\mu^{\pi_E}(c) := \text{MEAN}_{\tau \in \mathcal{D}}(c^\gamma(\tau))$
10:     **compute** soft loss $L_{\text{soft}}(c) := -J_\mu^{\pi_{mix}}(c) + \lambda \text{RELU}(J_\mu^{\pi_E}(c) - \beta)$
11:     **optimize** constraint function $c$: $\phi \leftarrow \phi - \eta_3 \nabla_\phi L_{\text{soft}}(c)$
12: **end for**
    **output:** constraint function $c$

---

**Baselines and metrics.** We use two baselines to compare against our method: (a) *GAIL-Constraint*, which is based on Generative adversarial imitation learning method (Ho & Ermon, 2016), and (b) *Inverse constrained reinforcement learning (ICRL)* (Malik et al., 2021), which is a recent method that can learn arbitrary neural network constraints. In the absence of any other neural network constraint learning technique, only those two relevant baselines are compared to empirically. We provide further details in Appendix C.2. Next, we define two metrics for our experiments: (a) *Constraint Mean Squared Error (CMSE)*, which is the mean squared error between the true constraint and the recovered constraint, and (b) *Normalized Accrual Dissimilarity (NAD)*, which is a dissimilarity measure computed between the expert and agent accruals (state-action visitations of policy/dataset). Further details can be found in Appendix C.3.

Finally, we use a constraint neural network with a sigmoid output, so that we can also interpret the outputs as safeness values (this is able to capture the true constraints arbitrarily closely, as justified in Appendix C.10). Our results are reported in Tables 1, 2, 5 and 6. The average recovered constraint functions and accruals are provided in Appendix D. For the highway datasets, our results are reported in Figures 2 and 11. The hyperparameter configuration (e.g., choice of $\lambda$), training strategy and training time statistics are elaborated in Appendix C.

## 5.1 RESULTS FOR SYNTHETIC EXPERIMENTS

*Results and discussion for the other environments can be found in Appendix B.6, B.7 and B.8.*

Table 1: Constraint Mean Squared Error (Mean ± Std. Deviation across 5 seeds)

| Algorithm↓, Environment→ | Gridworld (A) | Gridworld (B) | CartPole (MR) | CartPole (Mid) |
|---|---|---|---|---|
| GAIL-Constraint | 0.31 ± 0.01 | 0.25 ± 0.01 | 0.12 ± 0.03 | 0.25 ± 0.02 |
| ICRL | 0.11 ± 0.02 | 0.21 ± 0.04 | 0.21 ± 0.16 | 0.27 ± 0.03 |
| ICL (ours) | **0.08 ± 0.01** | **0.04 ± 0.01** | **0.02 ± 0.00** | **0.08 ± 0.05** |

Table 2: Normalized Accrual Dissimilarity (Mean ± Std. Deviation across 5 seeds)

| Algorithm↓, Environment→ | Gridworld (A) | Gridworld (B) | CartPole (MR) | CartPole (Mid) |
|---|---|---|---|---|
| GAIL-Constraint | 1.76 ± 0.25 | 1.29 ± 0.07 | 1.80 ± 0.24 | 7.23 ± 3.88 |
| ICRL | 1.73 ± 0.47 | 2.15 ± 0.92 | 12.32 ± 0.48 | 13.21 ± 1.81 |
| ICL (ours) | **0.36 ± 0.10** | **1.26 ± 0.62** | **1.63 ± 0.89** | **3.04 ± 1.93** |

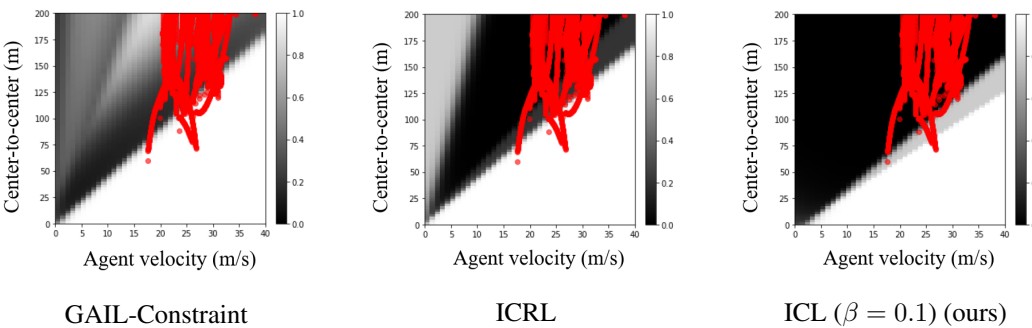

| GAIL-Constraint | ICRL | ICL ($\beta = 0.1$) (ours) |

Figure 2: Constraint functions (averaged across 5 seeds) recovered for the HighD highway driving environment. X-axis is the agent (ego car) velocity $v$ ($ms^{-1}$), Y-axis is the gap from the ego car to the vehicle (center-to-center distance) in front $g$ (m). Red points are (discretized) binary expert accruals, i.e. the states that the agent has been in. Both the baselines, GAIL-Constraint and ICRL assign a high (white) constraint value to more expert states compared to our method, ICL, which overlaps less with the expert states. Ideally, it is expected that the constraint value should be less for expert states, since they are acceptable states. Further discussion is provided in Appendix B.7.

**Accuracy and sharpness of learned constraints.** While our (practical) method is not guaranteed to produce the true constraint function (constraint unidentifiability is a known problem (Chou et al., 2020)), empirically, we find that our method is still able to learn constraint functions that strongly resemble the true constraint function, as can be seen by our low CMSE scores (Table 1). Other methods, e.g., GAIL-Constraint can find the correct constraint function for all environments except CartPole (Mid), however, the recovered constraint is more diffused throughout the state action space (see Appendix D for training plots). In contrast, our recovered constraint is pretty sharp, even without a regularizer.

**Ability to learn complex constraints.** From the training plots, we find that the ICRL method is able to find the correct constraint function only for CartPole (MR) and to a less acceptable degree for Gridworld (A). This is surprising as ICRL should be able to theoretically learn any arbitrary constraint function (note that we used the settings in (Malik et al., 2021) except for common hyperparameters), and expect it to perform better than GAIL-Constraint. Our justification for this is two-fold. One, the authors of ICRL have only demonstrated their approach with simple single-proposition constraints, and for more complex settings, ICRL may not be able to perform as well. Second, ICRL may require more careful hyperparameter tuning for each constraint function setting, even with the same environment, depending on the constraint. On the other hand, our method is able to learn these relatively complex constraints satisfactorily with relatively fewer hyperparameters.

**Similarity of learned policy to expert policy.** We find a strong resemblance between the accruals recovered by our method and the expert, as can be seen by our low NAD scores (Table 2). In these scores, there is some variability (e.g., in Cartpole (MR) and Gridworld (B) environments) which is expected, since the policies are stochastic. For baselines, GAIL-Constraint accruals are similar to the expert accruals except for CartPole (Mid) environment, where it is also unable to learn the correct constraint function. Overall, this indicates that GAIL is able to correctly imitate the constrained

expert across most environments, as one would expect. On the other hand, ICRL accruals are even worse than GAIL, indicating that it is unable to satisfactorily imitate the constrained expert, even on the environments for which it is able to generate a somewhat satisfactory constraint function. Training plots are provided in Appendix D.

## 5.2 Ablations and Questions

**Use of penalty method vs. Lagrangian.** As mentioned in Section 4.2, Lagrangian formulations are quite popular in the literature, however, they are inherently min-max problems that admit gradient ascent-descent solutions which are prone to oscillatory behavior, as we observe in Appendix B.4. Thus, we prefer the simpler framework of penalty method optimization which works well empirically.

**Stochastic dynamics.** Our algorithm can also be applied to stochastic environments, which we demonstrate in Appendix B.3 with slippery Gridworld environments. Stochasticity would lead to noisier demonstrations, and we find that a little noise in the demonstrations helps in the quality of the recovered constraint, but too much noise can hamper the constraint learning.

**Quality of learned policy w.r.t. imitation learning.** We reiterate that our objective is to learn constraints, while the goal of imitation learning techniques (e.g., behavior cloning) is to directly learn a policy that mimics the expert without inferring any reward or constraint function. Even then, once the constraints have been learned, they can be used with the given reward to learn a constrained policy. Depending on factors like quality of expert data etc., ICL can learn a better or worse policy compared to an imitation learning method. We do not have definitive evidence to conclude that either is true.

**Entropy regularized approaches and overfitting.** Maximum entropy techniques (e.g., (Scobee & Sastry, 2019; Malik et al., 2021)) have been previously proposed in the literature. Entropy maximization can indeed help in reducing overfitting, but they require a completely different mathematical setup. Our method doesn't belong to this family of approaches, but we hope to incorporate these strategies in the future to reap the mentioned benefits. We still note that empirically, our approach doesn't seem to suffer from any overfitting issues.

**Use of normalizing flow and reweighting.** We justify our choice of using a normalizing flow to reweigh policies and trajectories (Algorithm 3) in Appendix B.2. We find that ICL with reweighting performs better than ICL without reweighting, although the improvement is minor.

## 6 Conclusion, Limitations and Future Work

This paper introduces a new technique to learn (soft) constraints from expert demonstrations. While several techniques have been proposed to learn constraints in stochastic domains, they adapt the maximum entropy inverse RL framework to constraint learning (Scobee & Sastry, 2019; Park et al., 2020; McPherson et al., 2021; Glazier et al., 2021; Malik et al., 2021). We propose the first approach that can recover soft cumulative constraints common in constrained MDPs.

**Limitations.** First, the proposed technique assumes that the reward function is known and can only learn a single constraint function. The challenge with simultaneously learning the reward function as well as multiple constraints is that there can be many equivalent configurations of reward and constraint functions that can generate the same trajectories (unidentifiability (Ng et al., 2000)). Second, it is possible that expert demonstrations are sub-optimal. In that case, we hypothesize that our method would learn alternative constraints such that the provided demonstrations become optimal for these constraints. Finally, our approach requires several outer iterations of forward CRL and constraint adjustment, and as a result, it should in principle require more training time than the existing baselines.

**Future work.** First, for learning rewards and/or multiple constraints, we need a way to specify preference for specific combinations of reward and constraint functions. Second, it may be more desirable to express constraints not through expectations, but through probabilistic notions, e.g., a cumulative soft constraint which holds (i.e., is $\leq \beta$) with probability $\geq p$. Finally, we hope to extend our work to handle demonstrations from multiple experts and incorporate entropy regularization into our framework for its benefits.

## 6.1 Acknowledgements

Resources used in preparing this research at the University of Waterloo were provided by Huawei Canada, the province of Ontario and the government of Canada through CIFAR and companies sponsoring the Vector Institute.

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

APPENDIX

## A  PROOF OF THEOREM 1

We give a proof by contradiction under 3 assumptions. The 3 assumptions are:

(i) The reward function is non-negative.

(ii) The space of policies is finite.

(iii) There is a unique policy $\pi^E$ that achieves $J_\mu^{\pi^E}(r)$. In other words no other policy achieves the same expected cumulative reward as $\pi^E$.

Note that the first assumption is not restrictive since we can always shift a reward function by adding a sufficiently positive constant to the reward of all state-action pairs such that the new reward function is always non-negative. This new reward function is equivalent to the original one since it does not change the value ordering of policies.

We argue that the second assumption is reasonable in practice. Even if we consider what appears to be a continuous policy space because the parameterization of the policy space is continuous, in practice, parameters have a finite precision and therefore the space of policies is necessarily finite (albeit possibly very large).

Based on assumption (ii), it should be clear that the alternation of the optimization procedures in Equation (3) and Equation (4) will converge to a fixed point. Since there are finitely many policies, and we add a new policy to $\Pi$ at each iteration (until we reach the fixed point), in the worst case, the alternation will terminate once all policies have been added to $\Pi$.

When the algorithm converges to a fixed point, let $\pi^*$ be the optimal policy found by policy optimization in Equation (3) and $c^*$ be the optimal constraint function found by constraint adjustment in Equation (4). According to Equation (3), we know that $J_\mu^{\pi^*}(c^*) \leq \beta$. Similarly, we show that the objective in Equation (4) is less than or equal to $\beta$:

$$\max_c \min_{\pi \in \Pi} J_\mu^\pi(c) \text{ subject to } J_\mu^{\pi^E}(c) \leq \beta \tag{8}$$

$$= \min_{\pi \in \Pi} J_\mu^\pi(c^*) \qquad\qquad \text{since } c^* \text{ is the optimal solution of equation 4} \tag{9}$$

$$\leq \beta \qquad\qquad \text{since } \pi^* \in \Pi \text{ and } J_\mu^{\pi^*}(c^*) \leq \beta \tag{10}$$

Next, if we assume that $\pi^*$ is not equivalent to $\pi^E$, then we can show that the objective in Equation (4) is greater than $\beta$.

$$\max_c \min_{\pi \in \Pi} J_\mu^\pi(c) \text{ such that } J_\mu^{\pi^E}(c) \leq \beta \tag{11}$$

$$\geq \min_{\pi \in \Pi} J_\mu^\pi(\hat{c}) \qquad\qquad \text{where we choose } \hat{c}(s,a) = \frac{r(s,a)\beta}{J_\mu^{\pi^E}(r)} \tag{12}$$

$$= \min_{\pi \in \Pi} J_\mu^\pi(r)\beta / J_\mu^{\pi^E}(r) \qquad\qquad \text{by substituting } \hat{c} \text{ by its definition} \tag{13}$$

$$> J_\mu^{\pi^E}(r)\beta / J_\mu^{\pi^E}(r) = \beta \qquad \text{since } J_\mu^\pi(r) \geq J_\mu^{\pi^E}(r) \ \forall \pi \in \Pi \text{ according to equation 3}$$
$$\text{and } J_\mu^\pi(r) \neq J_\mu^{\pi^E}(r) \text{ by assumption (iii)} \tag{14}$$

The key step in the above derivation is Equation (12), where we choose a specific constraint function $\hat{c}$. Intuitively, this constraint function is selected to be parallel to the reward function while making sure that $J_\mu^{\pi^E}(c) \leq \beta$. We know that all policies in $\Pi$ achieve higher expected cumulative rewards than $\pi^E$. This follows from the fact that the optimization procedure in Equation (3) finds an optimal policy $\pi^*$ at each iteration and this optimal policy is not equivalent to $\pi^E$ based on our earlier assumption. So the policies in $\Pi$ must earn higher expected cumulative rewards than $\pi^E$. So if we select a constraint

function parallel to the reward function then the policies in $\Pi$ will have constraint values greater than the constraint value of $\pi^E$ and therefore they should be ruled out in the constrained policy optimization Equation (3).

Since the inequalities 8-10 contradict the inequalities 11-14, we conclude that $\pi^*$ must be equivalent to $\pi^E$. ∎

# B    ADDITIONAL DISCUSSION

## B.1    CONSTRAINED PPO

For the forward constrained RL procedure used in synthetic and HighD environments, we use a modification of the PPO algorithm (Schulman et al., 2017) that approximately ensures constraint feasibility. This modification is based on the penalty method mentioned in Section 4.2. This modified PPO works in two steps. First, we update the policy parameters to ensure constraint feasibility. Then, we update the policy parameters to optimize the PPO objective. If the policy $\pi$ is parameterized by $\boldsymbol{\theta}$, then these two steps are:

$$\text{FEASIBILITY PROJECTION: } \boldsymbol{\theta} \leftarrow \boldsymbol{\theta} - \eta_1 \nabla_{\boldsymbol{\theta}} \text{RELU}(J_\mu^\pi(c) - \beta)$$

$$\text{PPO UPDATE: } \boldsymbol{\theta} \leftarrow \boldsymbol{\theta} - \eta_2 \nabla_{\boldsymbol{\theta}} \text{PPO-LOSS}(\pi)$$

The gradient for the feasibility projection step can be computed just like policy gradient:

$$G_{t_0}(c) := \sum_{t=t_0}^{\infty} \gamma^{t-t_0} c(s_t, a_t) \tag{15}$$

$$\nabla_{\boldsymbol{\theta}} J_\mu^\pi(c) = E_{s_0 \sim \mu, a_t \sim \pi(\cdot|s_t), s_{t+1} \sim p(\cdot|s_t, a_t)} \left[ \sum_{t=0}^{\infty} G_t(c) \nabla_{\boldsymbol{\theta}} \log \pi(a_t|s_t) \right] \tag{16}$$

$$\nabla_{\boldsymbol{\theta}} \text{RELU}(J_\mu^\pi(c) - \beta) = \mathbf{I}(J_\mu^\pi(c) > \beta) \nabla_{\boldsymbol{\theta}} J_\mu^\pi(c) \tag{17}$$

The PPO-LOSS($\pi$) term is computed over a minibatch as follows (note that $A(s_t, a_t)$ is an advantage estimate). This is the same as the original paper (Schulman et al., 2017), except for an entropy term.

$$r_t := \frac{\pi(a_t|s_t; \boldsymbol{\theta})}{\pi(a_t|s_t; \boldsymbol{\theta}_{old})} \tag{18}$$

$$\text{PPO-LOSS}(\pi) := -\mathbb{E}_t \left[ \min(r_t, \text{clip}(r_t, 1 - \epsilon_{PPO}, 1 + \epsilon_{PPO})) A(s_t, a_t) \right] - \lambda_{ent} H(\pi) \tag{19}$$

This gradient can be computed through automatic differentiation.

Overall, we do not have a RELU term in the PPO update, since its gradient would interfere with the PPO update and cause difficulty in learning. Instead, the feasibility projection step tries to ensure feasibility of the solution before every PPO update. While this doesn't guarantee that feasibility will hold during and after the PPO update, we see empirically that feasibility approximately holds.

The feasibility projection step is comparable to a projection step, since it projects the solution into the feasibility region, although not necessarily the closest feasible point.

Our results are reported in Figures 3, 4.

## B.2    ABLATION: EFFECT OF POLICY MIXTURE AND REWEIGHTING

To assess the effect of having a policy mixture and reweighting, we conduct experiments with two more variants of ICL on all the synthetic environments:

- ICL wihout a mixture policy or reweighting (use the policy learned in constrained RL step instead of $\pi_{mix}$ to do constraint adjustment step)
- ICL with a mixture policy but with no reweighting (use uniform weights)

Our results are reported in Figure 5. We find that ICL without a mixture policy or reweighting is unable to perform as well as other methods with Gridworld (A) environment. Empirically, it is

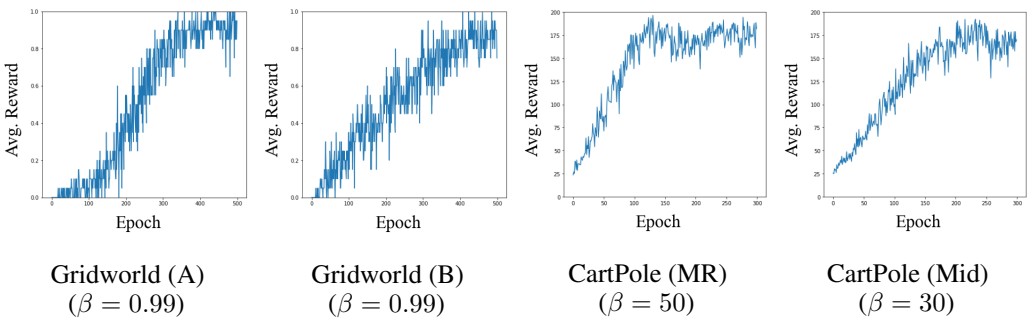

Figure 3: Avg reward per PPO epoch for the expert (1 seed). As seen, the modified PPO is able to learn a good policy that achieves a high reward.

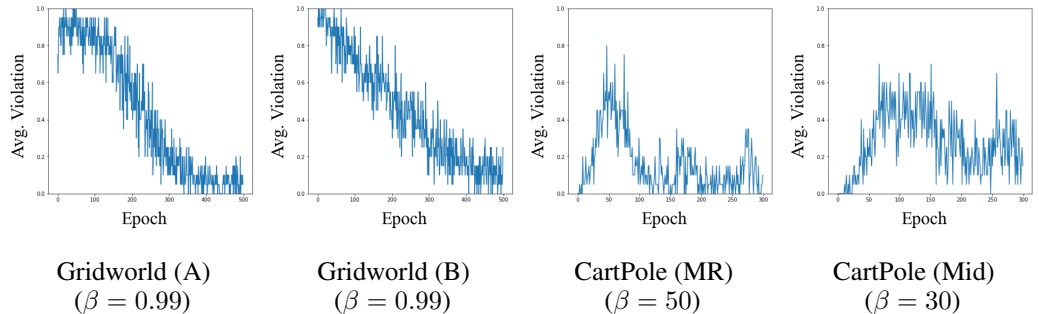

Figure 4: Avg constraint violations per PPO epoch for the expert (1 seed). As seen, the modified PPO is able to learn a good policy that reduces the constraint violations as the training progresses.

prone to divergence and cyclic behavior. This divergent behavior can be observed in the Gridworld (A) case. ICL with a mixture policy but with no reweighting is unable to converge as fast as ICL with mixture policy and reweighting in the Gridworld (B) and CartPole (Mid) environments. For the CartPole (MR) environment, all the methods converge fairly quickly and have almost the same performance. We also note that there is significant variance in the result for ICL on the CartPole (Mid) environment. Despite this, overall, ICL with a mixture policy and reweighting performs better than the other variants (or close to the best). Thus, we report this variant (with mixture policy and reweighting) in Table 1 and Table 2.

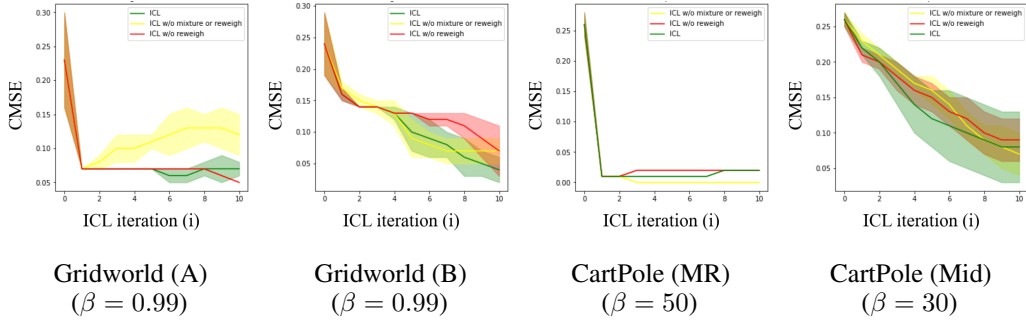

Figure 5: CMSE (Y-axis) vs ICL Iteration $i$ (X-axis). In each iteration, we do one complete procedure of constrained RL and one complete procedure of constraint function adjustment.

### B.3 ENVIRONMENT STOCHASTICITY

To assess the effect of environment stochasticity on the performance of our algorithm, we conduct experiments with a stochastic variant of Gridworld (A) environment. More specifically, we apply the

Table 3: Constraint Mean Squared Error

| Algorithm↓, Environment→ | $p_{slip} = 0.1$ | $p_{slip} = 0.3$ | $p_{slip} = 0.5$ |
|---|---|---|---|
| ICL | $0.02 \pm 0.00$ | $0.03 \pm 0.01$ | $0.08 \pm 0.00$ |

Table 4: Normalized Accrual Dissimilarity

| Algorithm↓, Environment→ | $p_{slip} = 0.1$ | $p_{slip} = 0.3$ | $p_{slip} = 0.5$ |
|---|---|---|---|
| ICL | $0.45 \pm 0.10$ | $1.13 \pm 0.37$ | $1.29 \pm 0.20$ |

Reported metrics (Mean $\pm$ Std. Deviation across 5 seeds) for the experiments on stochastic Gridworld (A) environments.

ICL algorithm (Algorithm 1) to the Gridworld (A) environment, with varying values of the transition probability. A transition probability of 1 implies deterministic transition to the next state. A transition probability of $1 - p_{slip}$ implies a stochastic transition, such that the agent can enter the intended next state with this probability, and go to any other random direction with overall probability $p_{slip}$. Our results are reported in Tables 3 and 4, and in Figure 6.

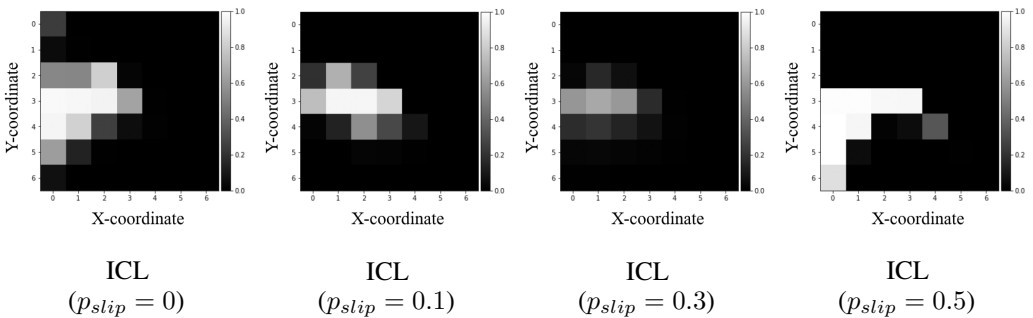

Figure 6: Average constraint function value (averaged across 5 training seeds) for stochastic Gridworld (A) environment, demonstrating the effect of changing $p_{slip}$.

Empirically, we observe that the increase in environment stochasticity leads to a decrease in the performance of the constrained RL algorithm. Additionally, with a higher stochasticity, there is more noise in the demonstrations. When we increase the stochasticity slightly (going from a deterministic environment to $p_{slip} = 0.1$), we find that the recovered constraint function has lower error (see Figure 6). This happens because with a little stochasticity, there is more exploration in terms of states visited, which leads to a more accurate constraint function since the algorithm tries to avoid more states. However, with more stochasticity ($p_{slip} = 0.5$), the demonstrations themselves become more noisy, and hence the algorithm recovers a worse constraint function than in the case with little stochasticity (see Table 3). Finally, we also find that accruals of the learned policy also have more error w.r.t. provided demonstrations as the stochasticity increases (see Table 4).

## B.4 Comparison w.r.t. a Lagrangian implementation of Equation (5)

Equation (5) can also be represented using a min-max optimization:

$$\min_{\lambda \geq 0} \max_{c} J_\mu^{\pi_{mix}}(c) - \lambda(J_\mu^{\pi_E}(c) - \beta) \tag{20}$$

It is possible to perform this optimization instead of the penalty based strategy proposed in this paper. In terms of implementation, this would amount to an alternating gradient ascent descent procedure, where we first ascent on $c$, followed by descent on $\lambda$, and we repeat these steps until convergence. However, as is the case with min-max optimizations, this is prone to training instability and oscillatory behavior. We demonstrate this with the Gridworld (A) environment. In particular, we can see this oscillatory behavior in the plot for the Lagrange multiplier $\lambda$ (Figure 7, second subfigure). We also

plot the expert satisfaction % for the Lagrangian and non Lagrangian implementation (ICL) (see third and fourth subfigure in Figure 7). The expert satisfaction % is computed as the percentage of expert demonstrations satisfying the constraint using the current constraint function, during training. As the plots show, the Lagrangian implementation is prone to oscillatory behavior. Initially it does achieve a high satisfaction, but over time, the satisfaction degrades. This does not happen in the ICL implementation. This is our primary rationale for using a penalty based method for Equation (5).

In terms of the recovered constraint function, we find that the Lagrangian implementation is still able to achieve a noisy yet reasonable constraint function (see the first subfigure in Figure 7). Therefore, the Lagrangian approach is indeed correct in principle and can be used to recover a constraint function. However, in terms of other training metrics, the Lagrangian approach does not perform well. Our proposed method is able to mitigate this.

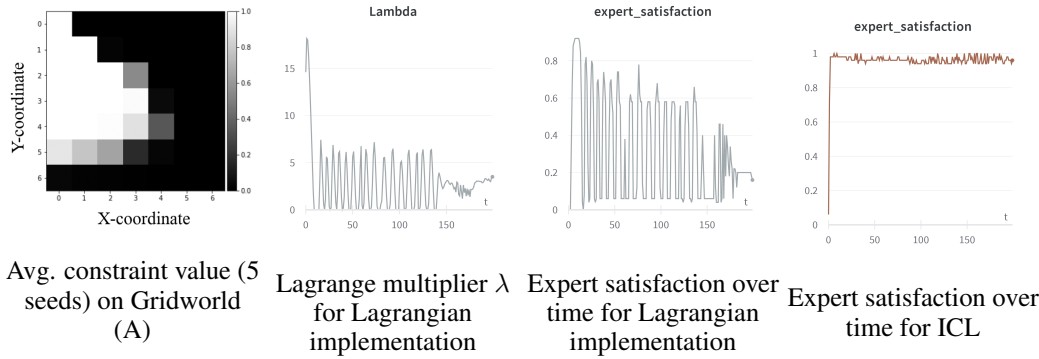

| Avg. constraint value (5 seeds) on Gridworld (A) | Lagrange multiplier $\lambda$ for Lagrangian implementation | Expert satisfaction over time for Lagrangian implementation | Expert satisfaction over time for ICL |

Figure 7: The first figure shows the avg. constraint function (averaged across 5 seeds) for the Lagrangian implementation. The second figure shows the value of the Lagrange multiplier during training for 1 seed (X-axis denotes the adjustment iteration). The last two figures show the value of the expert satisfaction (%) for the Lagrangian implementation and the regular implementation (ICL).

### B.5 EFFECT OF CONSTRAINT THRESHOLD $\beta$

We also assess the effect of $\beta$ with the Gridworld (A, B) environments. We do not assess the effect of $\beta$ on the CartPole (MR, Mid) environments since our claim can be easily verified by just observing the learned costs for the Gridworld (A, B) environments. Our claim is as follows. With a higher $\beta$, the learned constraint function should have a higher CMSE and more state-action pairs with higher constraint values. This is because since $\beta$ is higher, the agent is allowed to visit more high constraint value state-action pairs, as the constraint threshold is more relaxed. To verify this claim, we run experiments on the Gridworld (A, B) environments. Specifically, in addition to $\beta = 0.99$ (which corresponds to the results in Tables 1 and 2), we also run experiments with $\beta = 2.99, 5.99$. Our results are reported in Figures 8, 9. From the figures, it is apparent that our claim is correct.

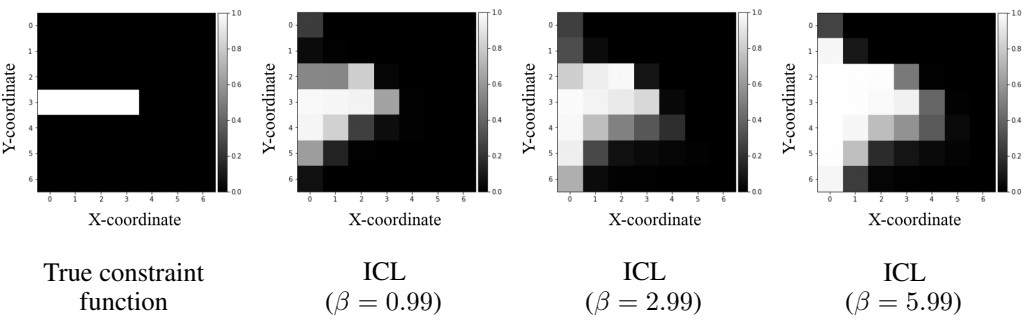

| True constraint function | ICL ($\beta = 0.99$) | ICL ($\beta = 2.99$) | ICL ($\beta = 5.99$) |

Figure 8: Average constraint function value (averaged across 5 training seeds) for Gridworld (A) environment, demonstrating the effect of changing $\beta$.

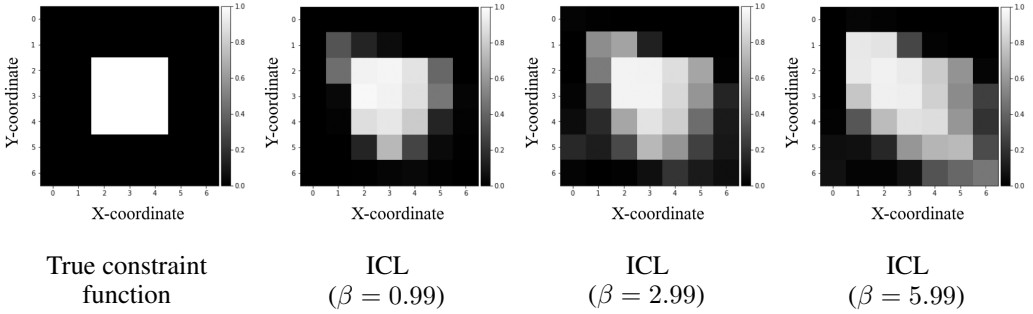

| True constraint function | ICL ($\beta = 0.99$) | ICL ($\beta = 2.99$) | ICL ($\beta = 5.99$) |

Figure 9: Average constraint function value (averaged across 5 training seeds) for Gridworld (B) environment, demonstrating the effect of changing $\beta$.

## B.6 MUJOCO EXPERIMENTS

We perform robotics experiments using the Mujoco (Todorov et al., 2012) environment setups described in the ICRL paper (Malik et al., 2021) (Ant and HalfCheetah environments). While there are four Mujoco environments in the ICRL paper, two of them use implicit constraints (Point and Ant-Broken environments). As we are learning constraint functions, we consider the two setups which use explicit constraints. To reduce training time, we use ICL with (highly optimized) PPO-Lagrange (Ray et al., 2019) instead of ICL with PPO-Penalty (Appendix B.1). *In principle, any procedure that can efficiently optimize Equation* (3) *can be used for the policy optimization part of ICL, as mentioned in Section 4.2.* The Mujoco environment setups are described in Appendix C.1 and the hyperparameter configuration for PPO-Lagrange is described in Table 11. Our results are reported in Tables 5 and 6.

Table 5: Constraint Mean Squared Error (Mean $\pm$ std. deviation across 5 seeds)

| Algorithm↓, Environment→ | Ant-Constrained | HalfCheetah-Constrained |
|---|---|---|
| GAIL-Constraint | $0.17 \pm 0.04$ | $0.20 \pm 0.03$ |
| ICRL | $0.41 \pm 0.00$ | $0.35 \pm 0.17$ |
| ICL (ours) | $\mathbf{0.07 \pm 0.00}$ | $\mathbf{0.05 \pm 0.00}$ |

Table 6: Normalized Accrual Dissimilarity (Mean $\pm$ std. deviation across 5 seeds)

| Algorithm↓, Environment→ | Ant-Constrained | HalfCheetah-Constrained |
|---|---|---|
| GAIL-Constraint | $8.02 \pm 2.84$ | $14.38 \pm 2.36$ |
| ICRL | $9.50 \pm 2.84$ | $\mathbf{7.50 \pm 4.97}$ |
| ICL (ours) | $\mathbf{6.84 \pm 1.29}$ | $10.16 \pm 7.49$ |

We observe that ICL is able to find the constraint more accurately compared to GAIL-Constraint and ICRL, as can be seen from the low CMSE scores in Table 5 (recovered constraint function plots in Figures 24 and 26). With respect to the accruals, our method achieves the lowest NAD score for Ant-Constrained environment (accrual plot in Figure 25). However, the NAD score achieved by ICL is not the lowest for HalfCheetah-Constrained environment, and instead, ICRL achieves a lower score (Table 6). In fact, ICRL succeeds at matching the accruals (low NAD score) for this environment better than ICL, although it finds a hard constraint.

This can be explained by visually inspecting the accruals for ICL and ICRL for the HalfCheetah-Constrained environment (Figure 27). More precisely, while both these accruals are shifted slightly to the right, which is appropriate for an agent trying to move right, ICRL has lower accrual mass for $z \leq 0$ and a larger accrual mass for $z \geq 2$, which indicates that it is able to find a better constrained optimal policy for this specific environment, compared to ICL (also see Figure 10). However, we do have a better constraint function (lower CMSE), and thus a constrained RL procedure that could find the optimal constrained policy in this specific setting could alleviate this issue.

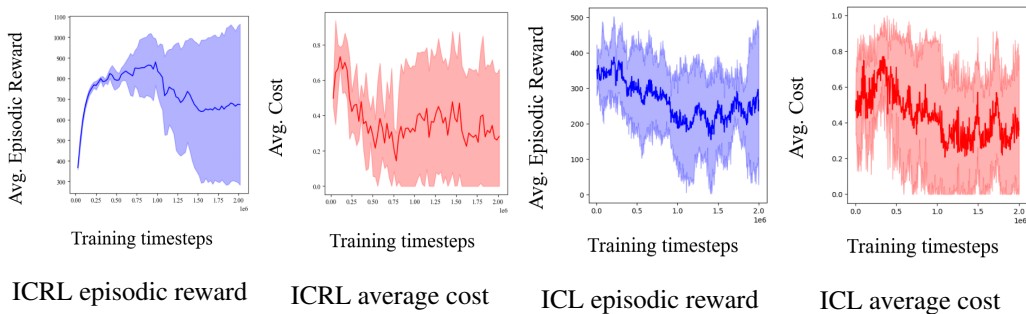

Figure 10: Training plots for 2M timesteps for ICRL and ICL.

For the Ant-Constrained environment, ICRL completely fails to recover a reasonable constraint (Figure 24). This is probably due to the challenging nature of the constraint. GAIL-Constraint performs moderately for these environments, while our method can find the (nearly) correct constraint.

### B.7 RESULTS FOR REAL WORLD EXPERIMENT (HIGHD)

We report the results for the HighD driving environment in Figure 2. Overall, all the methods are able to find a constraint function that shows that as the agent's velocity increases, the center-to-center distance that must be maintained between the cars also must increase. The time gap that must be maintained can be calculated by dividing the requisite center-to-center distance by the agent's velocity. For all the methods, this time gap is approximately between 3-4.5 seconds. More precisely, at $v = 20\text{ms}^{-1} = 72\text{kmh}^{-1}$, the gap is roughly between 60-80 m (equivalently, 3-4 seconds) for all methods, while close to $v = 40\text{ms}^{-1} = 144\text{kmh}^{-1}$, the gap is between 125-150 m (equivalently $\approx$3-4 seconds) for our method, and 160-180 m (equivalently 4-4.5 seconds) for GAIL-Constraint and ICRL. Note that our method is the only one which doesn't assign high constraint value to large center-to-center gaps (no white areas in the black regions in Figure 2). The possible justification for this is that the other methods are unable to explicitly ensure that expert trajectories are assigned low constraint value, while our method is able to do so through the constraint adjustment step. Moreover, our method has fewer accrual points (red) in the white (high constraint value) region compared to the baselines.

### B.8 RESULTS FOR REAL WORLD EXPERIMENT (EXID)

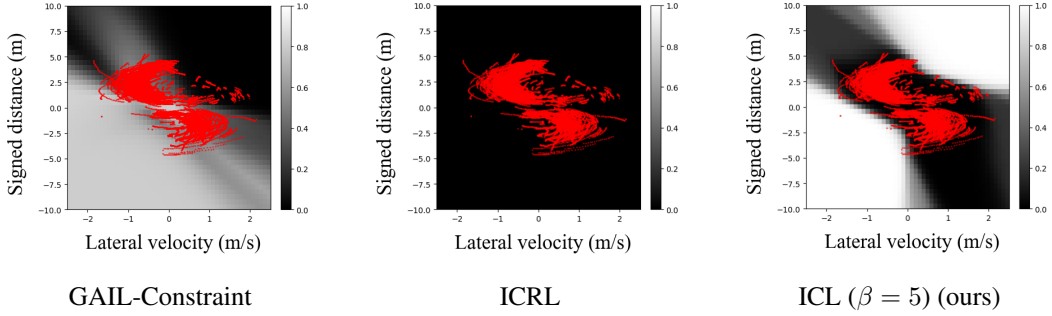

Figure 11: Constraint functions (averaged across 5 seeds) recovered for the ExiD highway lane change environment. White indicates a constraint value of 1, while black represents a constraint value of 0. X-axis is the agent (ego car) lateral velocity action $v$ ($ms^{-1}$), Y-axis is the signed distance $d$ ($m$) to the center line of the target lane. Red points are (discretized) binary expert accruals, i.e. the states/actions that the agent has been in. Both baselines, GAIL-Constraint and ICRL are unable to find a satisfactory constraint function for this task. On the other hand, our method, ICL, finds a reasonable constraint function.

We also conduct experiments with the ExiD environment (environment setup is described in Appendix C.1). The objective is to discover a constraint on the lateral velocity and signed distance to the target lane for doing a lane change in either direction. Note that just like the HighD environment, the real world constraint function is not known for this setting. The recovered constraint functions are reported in Figure 11 and the accruals are reported in Figure 29.

Overall, we find that only ICL is able to find a satisfactory constraint function for this setting. GAIL-Constraint finds a diffused constraint that overlaps the expert state-action pairs (and hence disallows them), while ICRL is unable to find a reasonable constraint. We can interpret the constraint plot of ICL as follows. Since the top-right and bottom-left quadrants ($v \geq 0, d \geq 0$ and $v \leq 0, d \leq 0$) are high in constraint value (white), they are relatively unsafe/prohibited. This is in line with expectation, since depending on the sign of the distance to the target lane, movement in one direction should be allowed and other should be disallowed (some risky behaviour is permitted since this is a soft constraint). More specifically, to reduce the magnitude of $d$, we must apply a control input $v$ of opposite sign. Further, the shape of the constraint is also not exactly square/rectangular. That is, it constrains movement to be in one direction when far away (e.g., for $d = 10$, $v \leq -1$), and allows movement in both directions as we get closer to the target lane (e.g., for $d = 2.5$, any $v$ is allowed).

## C  TRAINING SETUP AND EXPERIMENTAL DETAILS

All PPO variants mentioned in this paper are single-process for simplicity. For the tables, we abbreviate the Gridworld (A, B), CartPole (MR, Mid), Ant-Constrained, HalfCheetah-Constrained environments as GA, GB, CMR, CMid, Ant and HC.

### C.1  ENVIRONMENTS

We use the following environments in this work:

- **Gridworld (A, B)** are 7x7 gridworld environments (adapted from `yrlu`'s repository (Lu, 2019)). The start states, true constraint function and the goal state are indicated in Figure 12. The action space consists of 8 discrete actions including 4 nominal directions and 4 diagonal directions.

- **CartPole (MR, Mid)** (MR means Move Right) are variants of CartPole environment from OpenAI Gym (Brockman et al., 2016) where the objective is to balance a pole for as long as possible (maximum episode length is 200). The start regions and the true constraint function are indicated in Figure 12. Both variants may start in a region of high constraint value, and the objective is to move to a region of low constraint value and balance the pole there, while the constraint function is being learned.

- **HighD environment**: We construct an environment using $\approx 100$ trajectories of length $\leq 1000$ from the HighD highway driving dataset (Krajewski et al., 2018) (see Figure 13). The environment is adapted from the Wise-Move framework (Lee et al., 2019). For each trajectory, the agent starts on a straight road on the left side of the highway, and the objective is to reach the right side of the highway without colliding into any longitudinally moving cars. The state features consist of 7 ego features (x, y, speed, acceleration, rate of change of steering angle, steering angle and the heading angle) and 7 relevant predicates, propositions, and processed features (distance to the vehicle in front, processed distance, whether the ego is stopped, whether the ego has reached goal state, whether time limit has been exceeded, whether the ego is within road boundaries, and if ego has collided with another vehicle). The action space consists of a single continuous action, i.e., acceleration. Note that for this environment, we do not have the true constraint function. Instead, we aim to learn a constraint function that is able to capture the relationship between an agent's velocity and the distance to the car in front.

- **Ant-Constrained, HalfCheetah-Constrained environments** are *Mujoco* robotics environments used in (Malik et al., 2021) (see Figure 14). One difference in our experiments is the use of $z \geq -1$ constraint rather than $z \geq -3$ constraint used in the original paper,

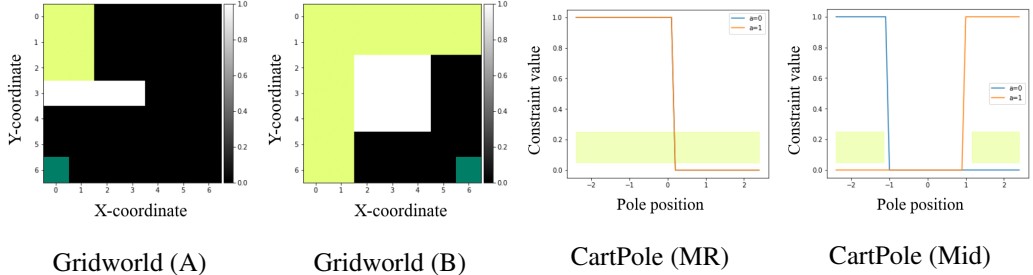

| Gridworld (A) | Gridworld (B) | CartPole (MR) | CartPole (Mid) |

Figure 12: Environments used in the experiments. White regions indicate constraint value of 1. Light green (or yellow) regions correspond to start states. Dark green regions correspond to the goal state. Gridworld environments have discrete states and discrete actions. Cartpole environments have continuous states and discrete actions. For the CartPole environments, the constraints represent desired behaviour. For CartPole (MR), the pole must stay in $x \geq 0.2$ and for CartPole (Mid), the pole is not allowed to go right ($a = 1$) for $x \geq 1$ and not allowed to go left ($a = 0$) for $x \leq -1$.

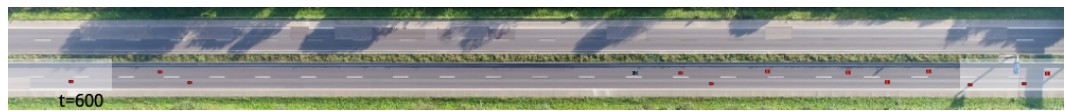

Figure 13: HighD driving dataset environment. Leftmost white region indicates start region and rightmost white region indicates end region. Ego car is in blue, other cars are in red. This environment has both discrete and continuous states, and a continuous action.

which is more challenging than the original setting. More concretely, the agent must stay in $z \in [-1, \infty)$ and it cannot go backwards, otherwise it will violate the constraint.

- **ExiD environment**: Similar to the HighD environment, we construct an environment using $\approx 1000$ trajectories of varying lengths ($\leq 1000$) from 5 different scenarios of ExiD highway driving dataset (Moers et al., 2022). To do so, we select trajectories where ego performs a lane change such that there are no other vehicles around the ego. On every reset, the environment initializes to the starting location of one of the $\approx 1000$ trajectories. The goal is to perform a lane change to reach the target lane, which can be either to the left or right. The state consists of the signed distance ($m$) to the target lane and the action is a continuous lateral velocity ($ms^{-1}$) that can be directly controlled by the agent.

The environments are more clearly illustrated in Figures 12 and 13.

## C.2 DESCRIPTION OF BASELINE METHODS

We use the following two baselines to compare against our method:

- **GAIL-Constraint**: Generative adversarial imitation learning (Ho & Ermon, 2016) is an imitation learning method that can be used to learn a policy that mimics the expert policy. The discriminator can be considered as a local reward function that incentivizes the agent to mimic the expert. We assume that the agent is maximizing the reward $r(s, a) := r_0(s, a) + \log(1 - c(s, a))$ where $r_0$ is the given true reward, and then the log term corresponds to the GAIL's discriminator. Note that when $c(s, a) = 0$, the discriminator reward is 0, and when $c(s, a) = 1$, the discriminator reward tends to $-\infty$.
- **Inverse constrained reinforcement learning (ICRL)** (Malik et al., 2021) is a recent method that is able to learn arbitrary Markovian neural network constraint functions, however, it can only handle hard constraints.

For both these methods, we use a similar training regime as adopted by (Malik et al., 2021), however, we try to keep the constraint function architecture fixed across all experiments (baselines and our method). *Note* that we did not include the method in Scobee and Sastry (Scobee & Sastry, 2019) as it is unable to handle continuous state spaces. Similarly, we did not include the binary constraint

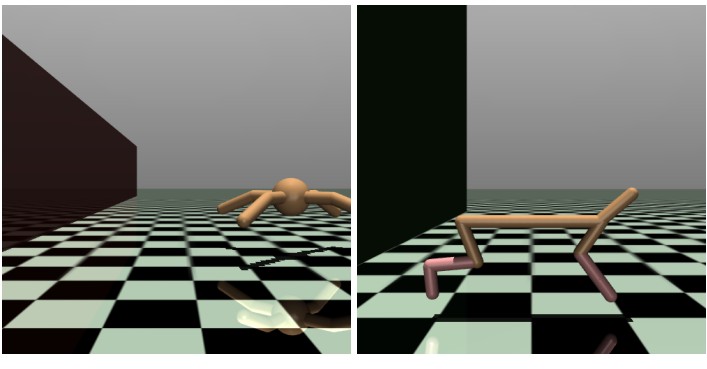

Ant-Constrained        HalfCheetah-Constrained

Figure 14: Mujoco environments used in the experiments. These environments have continuous states and continuous actions. Figures taken from ICRL paper (Malik et al., 2021). Instead of the original constraint $z \geq -3$, we consider a more challenging constraint $z \geq -1$.

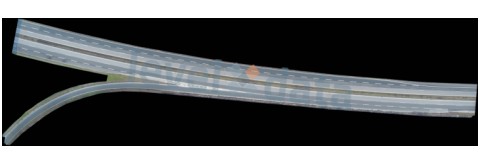

Real world map for the shown scenario.

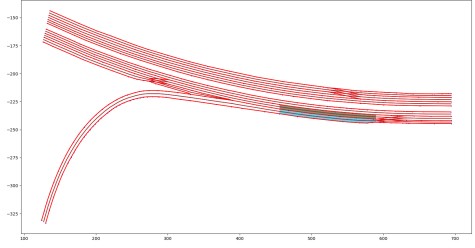

Start (cyan) and target (green) lanes for the lane change in this scenario.

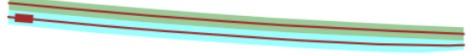

Simplified environment (agent starts from left-most point)

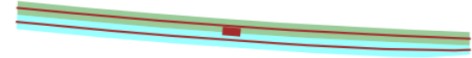

Simplified environment (agent moves according to chosen action)

Figure 15: ExiD dataset lane change environment (one of multiple scenarios). The state space contains the signed distance to the target lane, and the action space contains the lateral velocity. There are no other vehicles, only the ego car. The objective is to find a constraint on the lateral velocity (action) that an agent must maintain for any certain signed distance from the target lane.

(BC) method described in (Malik et al., 2021) as GC and ICRL empirically perform better than BC, according to their experiments.

## C.3 METRICS

We define two metrics for our experiments.

- **Constraint Mean Squared Error (CMSE)** is computed as the mean squared error between the true constraint function and the recovered constraint function on a uniformly discretized state-action space for the respective environment.
- **Normalized Accrual Dissimilarity (NAD)** is computed as follows. Given a policy learned by the method, we compute an agent dataset of trajectories. Then, the accrual (state-action visitation frequency) is computed for both the agent dataset and the expert dataset over a uniformly discretized state-action space, which is the same as the one used for CMSE. Finally, the accruals are normalized to sum to 1, and the Wasserstein distance is computed between the accruals.

More concretely, the NAD metric is computed as follows. Given the policy $\pi$ learned by any method (ICL or a baseline), we sample a dataset $\mathcal{D}_\pi$ of agent trajectories (note that $|\mathcal{D}_\pi| = |\mathcal{D}|$, where $\mathcal{D}$ is the expert dataset):

$$\mathcal{D}_\pi := \text{SAMPLE}_\tau(\Pi = \{\pi\}, p = \{1\}), \text{ SAMPLE is defined in Algorithm 1}$$

Then, the accrual (state-action visitation frequency) is computed for both the agent dataset and the expert dataset over a uniformly discretized state-action space (this depends on the constraint function inputs). For the various environments, these are:

- **Gridworld (A, B)**: $\{(x, y)|x, y \in \{0, 1, ..., 6\}\}$
- **CartPole (MR, Mid)**: $\{(x, a)|x \in \{-2.4, -2.3, \cdots, 2.4\}, a \in \{0, 1\}\}$
- **HighD**: $\{(v, g)|v \in \{0, 1, \cdots, 40\}, g \in \{0, 1, \cdots, 80\}\}$ (here, $v$ is in $ms^{-1}$ and $g$ is in pixels, where 1 pixel = 2.5m)
- **Ant-Constrained, HalfCheetah-Constrained**: $\{z|z \in \{-5, -4.9, \cdots, 5\}\}$
- **ExiD**: $\{(d, v)|d \in \{-10, -9.5, \cdots, 10\}, v \in \{-2.5, -2.4, \cdots, 2.5\}\}$ (here $d$ is in $m$ and $v$ is in $ms^{-1}$)

Finally, the accruals are normalized to sum to 1, and the Wasserstein distance is computed between the accruals. If the environment is CartPole/Ant-Constrained/HalfCheetah-Constrained, the 1D distance is computed and summed across actions. Otherwise, the 2D distance is computed. We use the Python Optimal Transport library (Flamary et al., 2021), specifically the `emd2` function to achieve this.

## C.4 COMMON HYPERPARAMETERS

Hyperparameters common to all the experiments are listed in Table 7.

Since HighD/ExiD/Ant-Constrained/HalfCheetah-Constrained are continuous action space environments, we use TANH activation for the policy, which outputs the mean and std. deviation of a learned Gaussian action distribution.

The inputs to the constraint functions are as follows:

- **Gridworld (A, B)**: $x, y$ coordinates of the agent's position in the 7x7 grid
- **CartPole (MR, Mid)**: $x$ position of the cart, and the discrete action (0/1 corresponding to left or right)
- **HighD**: velocity ($ms^{-1}$) of the ego, and the distance of the ego to the front vehicle in pixels (1 pixel $\approx 2.5\,m$)
- **Ant-Constrained, HalfCheetah-Constrained**: z-coordinate of the agent
- **ExiD**: signed distance ($m$) of the ego to the center line of the target lane, and the lateral velocity action ($ms^{-1}$) of the ego

Table 7: Common Hyperparameters

| Hyperparameter | Value(s) |
|---|---|
| PPO learning rate ($\eta_2$) | $5 \times 10^{-4}$ |
| Constraint function learning rate ($\eta_3$) | $5 \times 10^{-4}$ |
| Constraint function hidden layers | 64+RELU, 64+RELU |
| Constraint function final activation | SIGMOID |
| PPO policy layers | 64+RELU, 64+RELU |
| PPO value fn. layers | 64+RELU, 64+RELU |
| Minibatch size | 64 |
| PPO clip parameter ($\epsilon_{PPO}$) | 0.1 |
| PPO entropy coefficient ($\lambda_{ent}$) | 0.01 |
| PPO updates per epoch | 25 |
| No. of trajectories for any dataset | 50 |
| Training seeds | 1, 2, 3, 4, 5 |

The environment maximum horizons (episodes are terminated after these many steps) are as follows:

- **Gridworld (A, B)**: 50 steps
- **CartPole (MR, Mid)**: 200 steps
- **HighD**: 1000 steps
- **Ant-Constrained**: 500 steps
- **HalfCheetah-Constrained**: 1000 steps
- **ExiD**: 1000 steps

## C.5 HYPERPARAMETERS FOR GAIL-CONSTRAINT

Hyperparameters specific to the GAIL-Constraint method are listed in Table 8. This method was adapted from Malik et al. (Malik et al., 2021) and they performed 1M timesteps of training on all environments. To be sure that this method performs as expected, we did 2M timesteps of training (except for the HighD environment where we do 0.4M timesteps of training, since it was computationally expensive to simulate).

Table 8: GAIL-Constraint Hyperparameters

| Hyperparameter | Environment | | | | |
|---|---|---|---|---|---|
| | GA | GB | CMR | CMid | HighD |
| Discount factor ($\gamma$) | 1.0 | 1.0 | 0.99 | 0.99 | 0.99 |
| PPO steps per epoch | 2000 | 2000 | 2000 | 2000 | 2000 |
| PPO value fn. loss coefficient | 0.5 | 0.5 | 0.5 | 0.5 | 0.5 |
| PPO gradient clip value | 0.5 | 0.5 | 0.5 | 0.5 | 0.5 |
| PPO total steps | 2M | 2M | 2M | 2M | 0.4M |

| Hyperparameter | Environment | | |
|---|---|---|---|
| | Ant | HC | ExiD |
| Discount factor ($\gamma$) | 0.99 | 0.99 | 0.99 |
| PPO steps per epoch | 4000 | 4000 | 1000 |
| PPO value fn. loss coefficient | 0.5 | 0.5 | 0.5 |
| PPO gradient clip value | 0.5 | 0.5 | 0.5 |
| PPO total steps | 3.5M | 3.5M | 3.5M |

## C.6 HYPERPARAMETERS FOR ICRL

Similar to GAIL-Constraint, we performed 2M timesteps of training on all environments except HighD, where we performed 0.4M timesteps of training. Hyperparameters are listed in Table 9. ICRL

did not perform as expected on two out of the four synthetic environments, that is, Gridworld (B) and the CartPole (Mid) environments, and may require further environment specific hyperparameter tuning. Nevertheless, we use these parameters since they (somewhat) worked on Gridworld (A) and CartPole (MR) environments. Note that for a fair comparison, we tried to ensure that for some of the parameters, we use the same value as our method, ICL (these parameters are number of rollouts, batch size, learning rate, $\gamma$, constraint function architecture etc.).

Table 9: ICRL Hyperparameters

| Hyperparameter | Environment | | | | |
|---|---|---|---|---|---|
| | GA | GB | CMR | CMid | HighD |
| Discount factor ($\gamma$) | 1.0 | 1.0 | 0.99 | 0.99 | 0.99 |
| PPO steps per epoch | 2000 | 2000 | 2000 | 2000 | 2000 |
| PPO value fn. loss coefficient | 0.5 | 0.5 | 0.5 | 0.5 | 0.5 |
| Constraint value fn. loss coefficient | 0.5 | 0.5 | 0.5 | 0.5 | 0.5 |
| PPO gradient clip value | 0.5 | 0.5 | 0.5 | 0.5 | 0.5 |
| Eval episodes | 100 | 100 | 100 | 100 | 100 |
| ICRL Iterations | 200 | 200 | 200 | 200 | 40 |
| Forward PPO timesteps | 10000 | 10000 | 10000 | 10000 | 10000 |
| PPO Budget | 0 | 0 | 0 | 0 | 0 |
| Penalty initial value ($\nu$) | 0.1 | 0.1 | 0.1 | 0.1 | 0.1 |
| Penalty learning rate | 0.01 | 0.01 | 0.01 | 0.01 | 0.01 |
| Per step importance sampling | Yes | Yes | Yes | Yes | Yes |
| Forward KL (Old/New) | 10 | 10 | 10 | 10 | 10 |
| Backward KL (New/Old) | 2.5 | 2.5 | 2.5 | 2.5 | 2.5 |
| Backward iterations | 5 | 5 | 5 | 5 | 5 |
| Constraint network reg. coefficient | 0.6 | 0.6 | 0.6 | 0.6 | 0.6 |

| Hyperparameter | Environment | | |
|---|---|---|---|
| | Ant | HC | ExiD |
| Discount factor ($\gamma$) | 0.99 | 0.99 | 0.99 |
| PPO steps per epoch | 4000 | 4000 | 1000 |
| PPO value fn. loss coefficient | 0.5 | 0.5 | 0.5 |
| Constraint value fn. loss coefficient | 0.5 | 0.5 | 0.5 |
| PPO gradient clip value | 0.5 | 0.5 | 0.5 |
| Eval episodes | 100 | 100 | 100 |
| ICRL Iterations | 125 | 125 | 250 |
| Forward PPO timesteps | 10000 | 10000 | 10000 |
| PPO Budget | 0 | 0 | 0 |
| Penalty initial value ($\nu$) | 0.1 | 0.1 | 0.1 |
| Penalty learning rate | 0.01 | 0.01 | 0.01 |
| Per step importance sampling | Yes | Yes | Yes |
| Forward KL (Old/New) | 10 | 10 | 10 |
| Backward KL (New/Old) | 2.5 | 2.5 | 2.5 |
| Backward iterations | 5 | 5 | 5 |
| Constraint network reg. coefficient | 0.6 | 0.6 | 0.6 |

## C.7 HYPERPARAMETERS FOR ICL (OUR METHOD)

The hyperparameters for our method are listed in Table 10. Our PPO implementation differs from the baselines (the baseline code was adapted from Malik et al. (Malik et al., 2021) official repository) as it does a fixed number of episodes per epoch rather than doing a fixed number of environment steps per epoch. We adjusted the number of PPO epochs in ICL so that each PPO procedure is roughly the same length (or less) as the ICRL/GAIL PPO procedure. Even then, the entire ICL training process requires a lot more steps than the GAIL-Constraint/ICRL baselines, since every iteration requires a complete PPO procedure. We have mentioned this in our conclusions section.

For the Gridworld (A, B) environments, we chose $\beta = 0.99$ to disallow any state of constraint value 1. For the CartPole (MR, Mid) environments, the cartpole may start in a high constraint value region.

To allow it to navigate to a low constraint value region, we chose a high $\beta$. We chose $\beta$ by doing a hyperparameter search on $\beta \in \{30, 50\}$ for the CartPole environments.

For some environments, we use a variant of ICL that uses PPO-Lagrange (Ray et al., 2019) as the forward constrained RL procedure. This implementation (provided by OpenAI) is highly optimized for constrained RL tasks, especially robotics tasks. We use this variant of ICL for our Mujoco and ExiD experiments. For this implementation, we mostly use the default parameters provided by Ray et al. (2019). A list of relevant hyperparameters are provided in Table 11. Certain hyperparameters have not been altered (e.g., learning rate, training iterations etc.). PPO epochs for the intermediate CRL procedures have also been kept at 50 (as in the source code), which is sufficient to achieve decent performance for these constrained environments. $\beta$ was chosen by doing a hyperparameter search on $\beta \in \{5, 15\}$ for the Mujoco and ExiD environments.

To choose $\lambda$, we tried values 1.5, 15, 150 and found that 1.5 was insufficient for the optimization. We know that $\lambda$ is the RELU multiplier term in the soft loss objective, and it cannot be a low value for constraint adjustment procedure. Hence, we chose a value of 15 which was appropriate for our experiments.

Table 10: ICL Hyperparameters

| Hyperparameter | Environment) | | | | |
|---|---|---|---|---|---|
| | GA | GB | CMR | CMid | HighD |
| Discount factor ($\gamma$) | 1.0 | 1.0 | 0.99 | 0.99 | 0.99 |
| ICL Iterations ($n$) | 10 | 10 | 10 | 10 | 3 |
| Correction learning rate ($\eta_1$) | $2.5 \times 10^{-5}$ | $2.5 \times 10^{-5}$ | $2.5 \times 10^{-5}$ | $2.5 \times 10^{-5}$ | $2.5 \times 10^{-5}$ |
| PPO episodes per epoch | 20 | 20 | 20 | 20 | 20 |
| $\beta$ | 0.99 | 0.99 | 50 | 30 | 0.1 |
| Constraint update epochs ($e$) | 20 | 20 | 20 | 20 | 25 |
| PPO epochs ($m$) | 500 | 500 | 300 | 300 | 50 |
| Soft loss coefficient $\lambda$ | 15 | 15 | 15 | 15 | 15 |

Table 11: ICL Hyperparameters (with PPO-Lagrange from Ray et al. (2019))

| Hyperparameter | Environment | | |
|---|---|---|---|
| | Ant | HC | ExiD |
| Discount factor ($\gamma$) | 0.99 | 0.99 | 0.99 |
| GAE Lambda ($\lambda_{GAE}$) | 0.97 | 0.97 | 0.97 |
| ICL Iterations ($n$) | 5 | 5 | 5 |
| PPO steps per epoch | 4000 | 4000 | 1000 |
| $\beta$ | 5 | 15 | 5 |
| Constraint update epochs ($e$) | 25 | 25 | 25 |
| PPO epochs ($m$) | 50 | 50 | 50 |

## C.8 TRAINING TIME STATISTICS

Training time statistics are provided in Table 12. For each environment and method, we report the average training time in hours and minutes, averaged across 5 seeds. In principle, if one forward constrained RL procedure takes $t$ time and one constraint adjustment procedure takes $t'$ time, we expect GAIL-Constraint and ICRL to take at least $2t$ time, since we run these procedures at least twice as long as one forward procedure (longer for Mujoco and ExiD environments, but still a constant times $t$). With $n$ iterations of ICL, ICL should take at least $n(t + t')$ time which depends on $t, t', n$. Empirically, since $n \leq 10$ and the implementation of forward constrained RL (PPO-Lagrange/PPO-Penalty) differs from the baselines, we observe that ICL is actually comparable to the baselines in overall runtime. We also note that the training times may depend on CPU/GPU load.

Table 12: Average training times (hours/minutes), averaged across 5 seeds. PPOPen refers to ICL implementation with PPO-Penalty, and PPOLag refers to the ICL implementation with PPO-Lagrange (Ray et al., 2019).

| Environment↓, Method→ | Average training time | | |
|---|---|---|---|
| | GAIL-Constraint | ICRL | ICL (n, type) |
| GA | 7 h 15 m | 10 h 43 m | 1 h 16 m (10, PPOPen) |
| GB | 7 h 13 m | 10 h 37 m | 1 h 20 m (10, PPOPen) |
| CMR | 7 h 39 m | 15 h 56 m | 9 h 43 m (10, PPOPen) |
| CMid | 8 h 0 m | 16 h 0 m | 7 h 40 m (10, PPOPen) |
| HighD | 1 h 47 m | 3 h 34 m | 5 h 9 m (3, PPOPen) |
| Ant | 2 h 8 m | 6 h 48 m | 6 h 25 m (5, PPOLag) |
| HC | 2 h 32 m | 7 h 28 m | 9 h 32 m (5, PPOLag) |
| ExiD | 1 h 52 m | 32 h 20 m | 2 h 22 m (5, PPOLag) |

### C.9 Expert dataset generation process

For synthetic experiments (Gridworld and CartPole) and robotics experiments (Mujoco), we simply perform constrained RL for sufficient number of epochs until convergence. Afterwards, we generate trajectories using the learned policy, ensuring that the expected discounted constraint value across the expert dataset is $\leq \beta$ for the specific environment.

For the HighD dataset, the data collection process is described in Krajewski et al. (2018). For this dataset, we choose one of the multiple scenarios in this dataset and use $\approx 100$ trajectories from this scenario that go from the start region to the end region. For the ExiD dataset, the data collection process is similarly described in Moers et al. (2022). For this dataset, we randomly choose 5 scenarios, and filter tracks to get $\approx 1000$ trajectories in each of which a vehicle performs a single lane change, with no vehicle in the target lane.

### C.10 Constraint function architecture choice

We also further assess whether the chosen constraint function architecture is sufficient to capture the true constraint arbitrarily closely. To perform this assessment, we consider the following constraint function architectures:

- architectures where the number of hidden nodes vary:
  - $(A_1)$: two hidden layers with 32 nodes each and RELU activation, followed by sigmoid activation for the output
  - $(B)$: two hidden layers with 64 nodes each and RELU activation, followed by sigmoid activation for the output; *this is the architecture we use for all the ICL and baseline experiments*
  - $(C_1)$: two hidden layers with 128 nodes each and RELU activation, followed by sigmoid activation for the output
- architectures where the number of hidden layers vary:
  - $(A_2)$: one hidden layers with 64 nodes and RELU activation, followed by sigmoid activation for the output
  - $(B)$: two hidden layers with 64 nodes each and RELU activation, followed by sigmoid activation for the output; *this is the architecture we use for all the ICL and baseline experiments*
  - $(C_2)$: three hidden layers with 64 nodes each and RELU activation, followed by sigmoid activation for the output

Given a particular constraint function architecture, we train to minimize error w.r.t. the known true constraint function, using supervised learning. This produces a lower bound on the CMSE for that setting. Intuitively, an architecture with more representational power will lead to a lower value of this lower bound. Our results are reported in Tables 13 and 14. As can be seen from the tables, having too few hidden nodes or layers is not sufficient to arbitrarily approximate the true constraint function.

Table 13: CMSE achieved by various constraint function architectures (varying number of hidden nodes), which can be treated as lower bounds for CMSE in the requisite settings.

| Environment↓, CMSE→ | Architecture | | |
|---|---|---|---|
| | $A_1$ | $B$ | $C_1$ |
| GA | $0.03 \pm 0.01$ | $0.00 \pm 0.00$ | $0.00 \pm 0.00$ |
| GB | $0.02 \pm 0.01$ | $0.00 \pm 0.00$ | $0.00 \pm 0.00$ |
| CMR | $0.00 \pm 0.00$ | $0.00 \pm 0.00$ | $0.00 \pm 0.00$ |
| CMid | $0.01 \pm 0.00$ | $0.00 \pm 0.00$ | $0.00 \pm 0.00$ |
| Ant | $0.00 \pm 0.00$ | $0.00 \pm 0.00$ | $0.00 \pm 0.00$ |
| HC | $0.00 \pm 0.00$ | $0.00 \pm 0.00$ | $0.00 \pm 0.00$ |

Table 14: CMSE achieved by various constraint function architectures (varying number of hidden layers), which can be treated as lower bounds for CMSE in the requisite settings.

| Environment↓, CMSE→ | Architecture | | |
|---|---|---|---|
| | $A_2$ | $B$ | $C_2$ |
| GA | $0.06 \pm 0.00$ | $0.00 \pm 0.00$ | $0.00 \pm 0.00$ |
| GB | $0.09 \pm 0.01$ | $0.00 \pm 0.00$ | $0.00 \pm 0.00$ |
| CMR | $0.01 \pm 0.00$ | $0.00 \pm 0.00$ | $0.00 \pm 0.00$ |
| CMid | $0.02 \pm 0.00$ | $0.00 \pm 0.00$ | $0.00 \pm 0.00$ |
| Ant | $0.01 \pm 0.00$ | $0.00 \pm 0.00$ | $0.00 \pm 0.00$ |
| HC | $0.01 \pm 0.00$ | $0.00 \pm 0.00$ | $0.00 \pm 0.00$ |

## D  TRAINING PLOTS

For completeness, we provide the training plots for the recovered constraint functions (Figures 16, 18, 20, 22, 24, 26) and their respective accruals (Figures 17, 19, 21, 23, 28, 25, 27, 29). Note that we have reported the costs for the HighD/ExiD environments in the main paper and in the Appendix (Figures 2, 11).

Please note that we have omitted the individual reward and constraint value plots, since there aren't any interesting insights. Typically, the reward goes up over time and the constraint value goes down over time, similar to Figures 3 and 4.

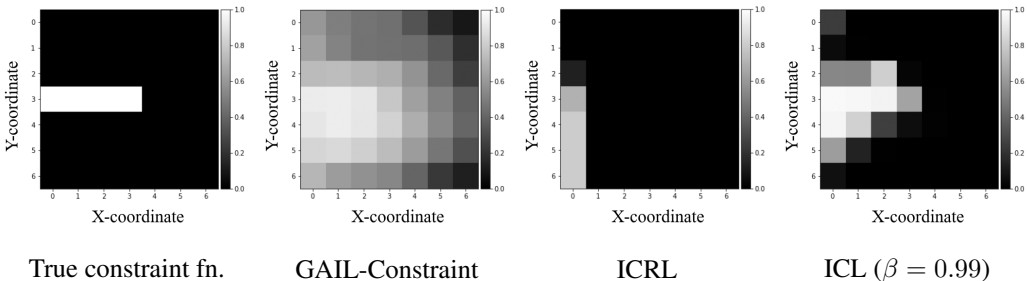

|  True constraint fn. | GAIL-Constraint | ICRL | ICL ($\beta = 0.99$) |

Figure 16: Average constraint function value (averaged across 5 training seeds) for Gridworld (A) environment.

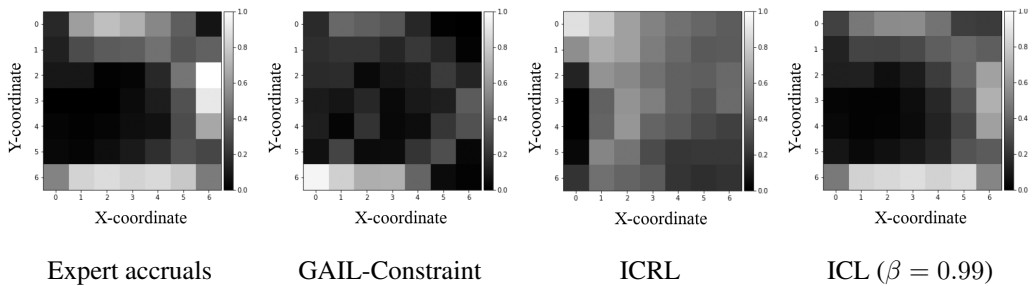

Figure 17: Average normalized accruals (averaged across 5 training seeds) for Gridworld (A) environment.

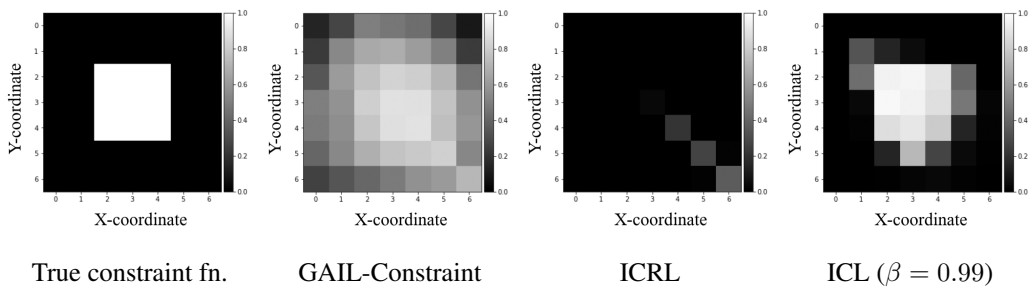

Figure 18: Average constraint function value (averaged across 5 training seeds) for Gridworld (B) environment.

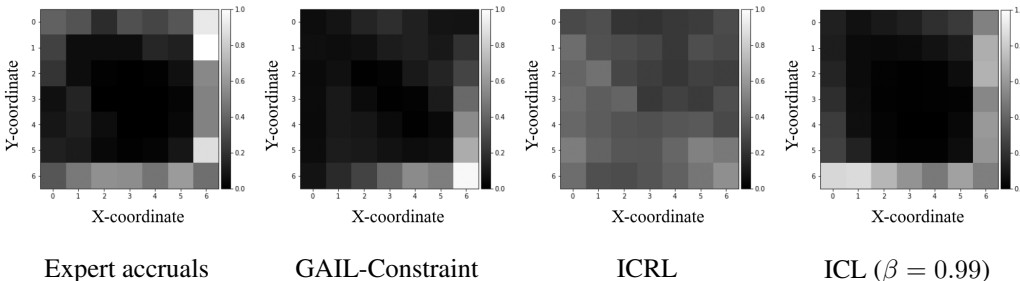

Figure 19: Average normalized accruals (averaged across 5 training seeds) for Gridworld (B) environment.

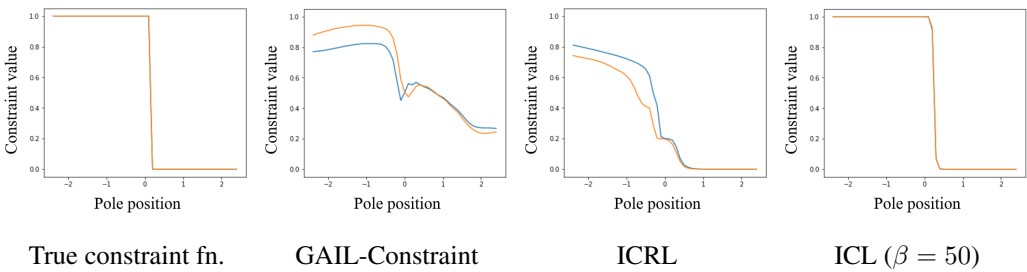

Figure 20: Average constraint function value (averaged across 5 training seeds) for CartPole (MR) environment. Blue line represents the constraint value for $a = 0$ (going left) and the orange line represents the constraint value for $a = 1$ (going right).

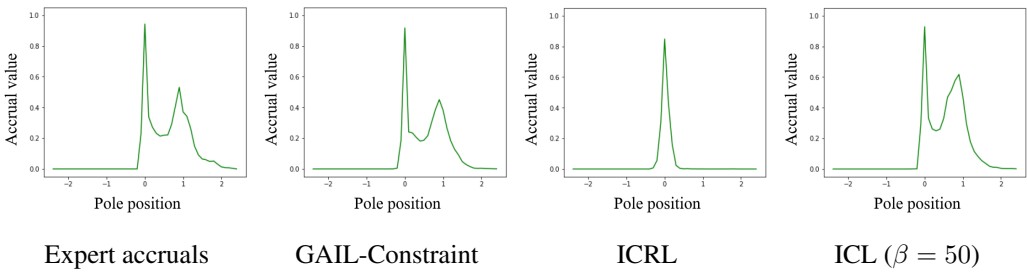

Figure 21: Average normalized accruals (averaged across 5 training seeds) for CartPole (MR) environment.

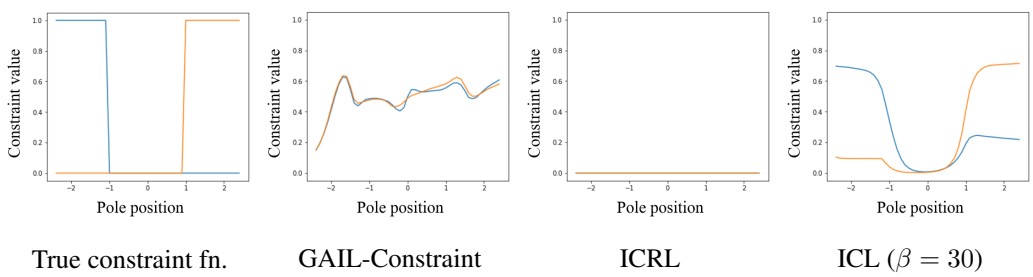

Figure 22: Average constraint function value (averaged across 5 training seeds) for CartPole (Mid) environment. Blue line represents the constraint value for $a = 0$ (going left) and the orange line represents the constraint value for $a = 1$ (going right).

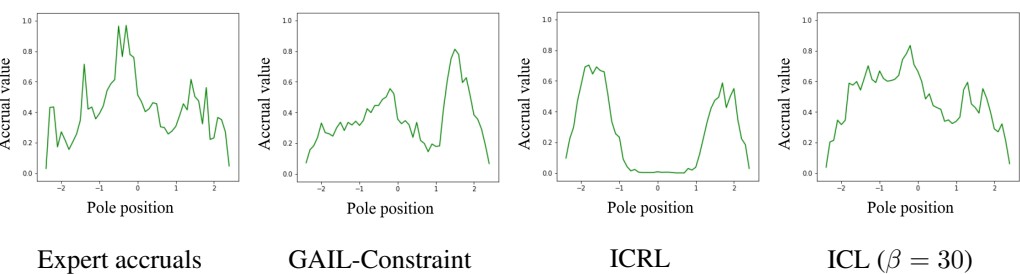

Figure 23: Average normalized accruals (averaged across 5 training seeds) for CartPole (Mid) environment.

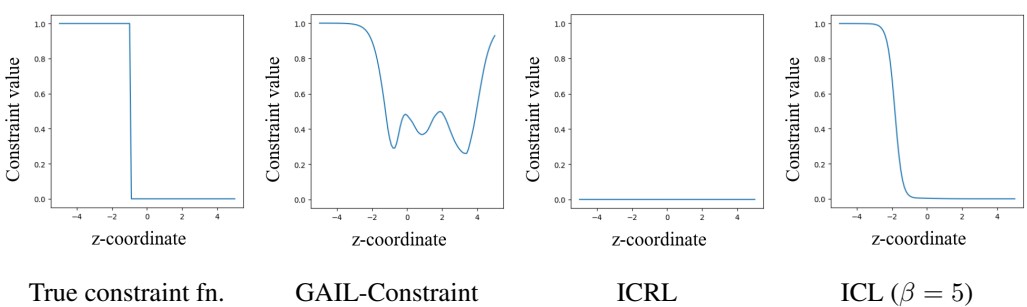

Figure 24: Average constraint function value (averaged across 5 training seeds) for Ant-Constrained environment.

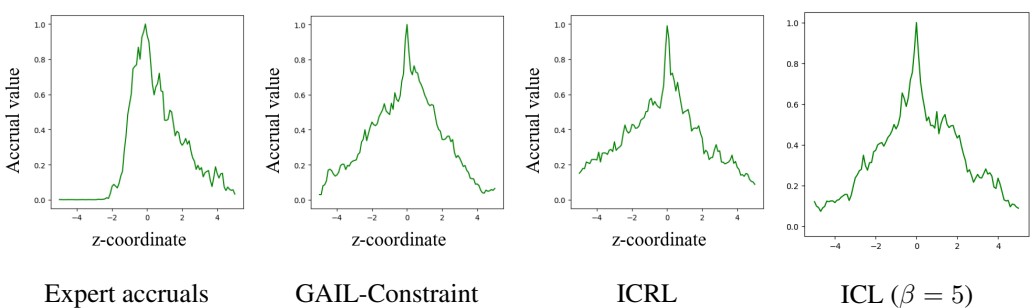

Figure 25: Average normalized accruals (averaged across 5 training seeds) for Ant-Constrained environment.

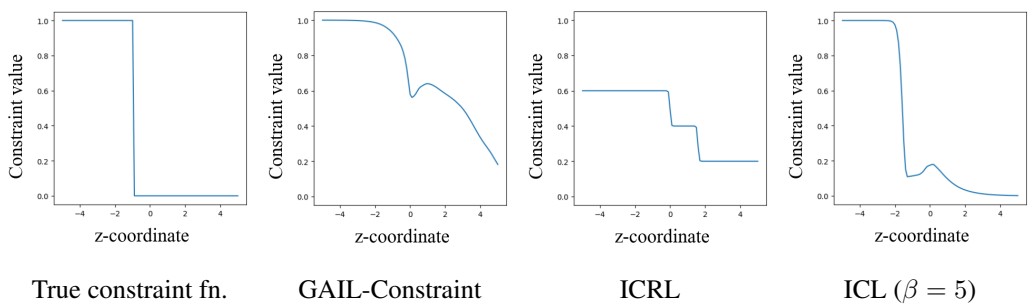

Figure 26: Average constraint function value (averaged across 5 training seeds) for HalfCheetah-Constrained environment.

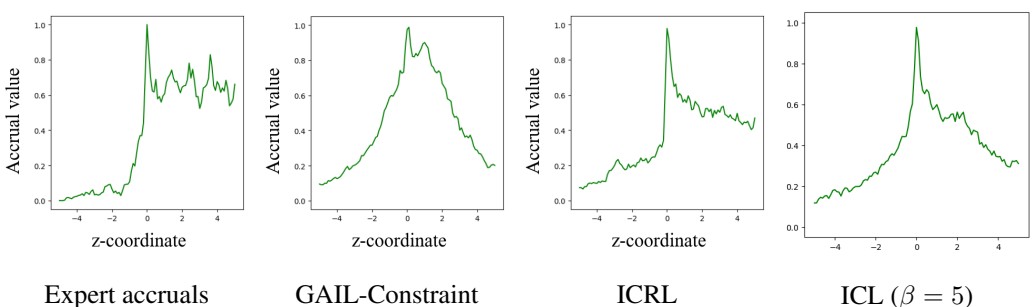

Figure 27: Average normalized accruals (averaged across 5 training seeds) for HalfCheetah-Constrained environment.

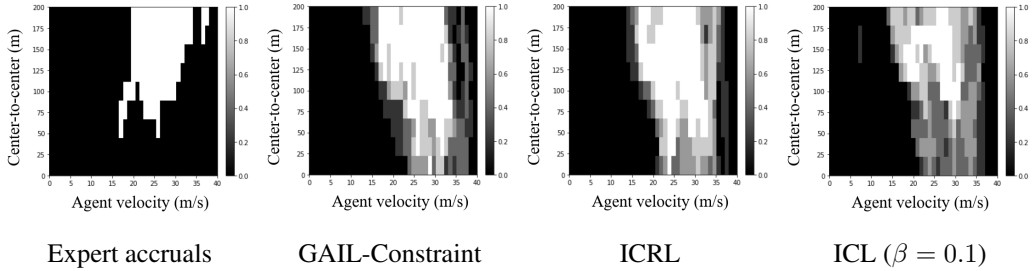

Figure 28: Average normalized binary accruals (averaged across 5 training seeds) for HighD environment. The X-axis and Y-axis are ego velocity in $ms^{-1}$ and gap to the front vehicle $g$ in $m$ respectively.

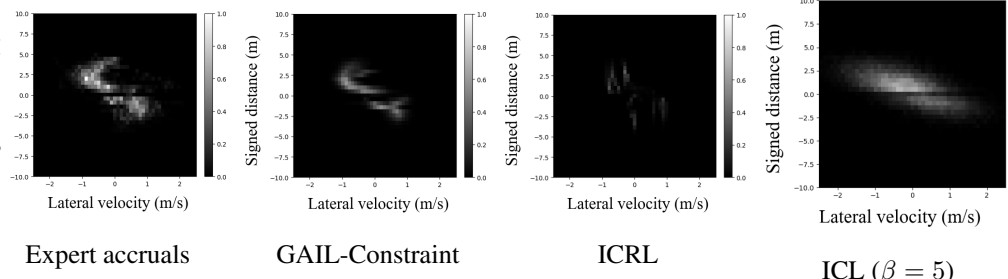

Figure 29: Average normalized accruals (averaged across 5 training seeds) for ExiD lane change environment. The X-axis and Y-axis are lateral velocity $v$ in $ms^{-1}$ and signed distance to the the center line of the target lane $d$ in $m$ respectively.

