# OpenReview forum: "Learning Soft Constraints From Constrained Expert Demonstrations"
_ICLR.cc/2023/Conference — ICLR 2023 notable top 25%_

### Official Review · Reviewer_5hKt · 2022-10-21

**Confidence:** 3
**Correctness:** 4
**Technical Novelty And Significance:** 3
**Empirical Novelty And Significance:** Not applicable
**Recommendation:** 6

**Clarity, Quality, Novelty And Reproducibility:**

**[Clarity & Quality]**
- The paper is mostly well-written and easy to understand.
- Detailed descriptions of the expert demonstrations used for their experiments are lacking.
- It would be better to include the experimental details (in Appendix C.1) in the main paper.

**[Novelty]**
- The proposed method is novel to my knowledge.

**[Reproducibility]**
- The authors have released their code.

**Strength And Weaknesses:**

**[Strengths]**
- The paper is well-written and easy to understand.
- The proposed approach is theoretically well-motivated and sensible.
- The experiments include ablation studies.

**[Weakness]**
- The main weakness of the paper is its limited empirical evaluation. The authors tested their method only on two toy environments (GridWorld and CartPole) and the HighD environment, which also seems relatively simple as it has a 1D action space (and presumably a 2D state space). Given the complexity of the method, it would be desirable to include more extensive comparisons on diverse, more challenging domains (e.g., constrained MuJoCo environments as in the ICRL paper).

**[Questions]**
- Is the constraint function $c$ in the paper a per-step function (i.e., $c(s_t, a_t)$) or a per-trajectory function ($c(\tau)$)? If the former is the case, there should be infinitely many ways to assign per-step constraints to the expert trajectories because the only constraint is that their sum should be bounded by $\beta$. How does the method deal with this?
- I didn't understand the CartPole figures in Fig. 10. What do the "a=0" and "a=1" lines mean? What are the axes?
- How is the state space defined in the HighD environment? How does the agent observe the cars in front of it? How are the expert data collected?

**Summary Of The Paper:**

The paper addresses the problem of inverse constraint learning, whose goal is to learn the unknown (soft) constraints that the expert demonstrations obey in a given expert dataset, assuming that the reward function is known. The authors propose a novel method named ICL, which learns a constraint function in a way that the total constraint is bounded for the expert demonstrations but maximized for agent behaviors. They demonstrate that their proposed method outperforms two baselines on three environments.

**Summary Of The Review:**

While their proposed method is well-motivated and sensible, I cannot recommend acceptance in its current form due to its limited experimental evaluation. I would be happy to raise my score if the authors address my concerns above.

-------------(11/16 Update)-------------

I raised my score from 5 to 6 as the response addressed my initial concerns.

---

> ### Author Response · Authors · 2022-11-16
> **Response 1**
>
> > **The main weakness of the paper is its limited empirical evaluation. The authors tested their method only on two toy environments (GridWorld and CartPole) and the HighD environment, which also seems relatively simple as it has a 1D action space (and presumably a 2D state space). Given the complexity of the method, it would be desirable to include more extensive comparisons on diverse, more challenging domains (e.g., constrained MuJoCo environments as in the ICRL paper).**
>
> As requested, we have conducted experiments with Ant and HalfCheetah environments, similar to ICRL paper [2]. Additionally, we also conducted experiments with the ExiD dataset (constructed using ~1000 trajectories) by finding a constraint for lane change in highway driving. The corresponding experimental details are available in Appendix C, and the constraint/accrual plots are available in Appendix D. The discussion is available in Appendix B.6 and B.8. We hope these experiments can address the stated concerns.
>
> > **Is the constraint function c in the paper a per-step function (i.e., c(s_t, a_t)) or a per-trajectory function (c(\tau))? If the former is the case, there should be infinitely many ways to assign per-step constraints to the expert trajectories because the only constraint is that their sum should be bounded by \beta. How does the method deal with this?**
>
> The constraint function c used in the paper is a per-step function (c(s_t, a_t)) which is standard in safe/constrained RL literature [1]. At first glance, it may seem that there are infinitely many ways to assign per-step constraints to the expert trajectories such that the discounted episodic cost is bounded by β. However, the optimality of expert trajectories implies that not all such assignments are valid, since only a few of such assignments will produce the expert demonstrations when constrained RL is performed with the given reward and the chosen constraint function assignment. We explain this requirement in Section 2, paragraph 3. There still can be multiple equivalent constraints that produce the same set of optimal constrained policies (unidentifiability is a known problem), which we acknowledge in Section 6. As explained in our response to reviewer EuC1, we try to reduce this unidentifiability by choosing a specific (canonical) architecture.
>
> More generally, per-trajectory constraint functions c(\tau) are a superset of per-step constraint functions c(s_t, a_t), since any per-step constraint can be represented using a per-trajectory constraint function, but the converse is not true because the per-trajectory constraint function may not always be decomposable into per-step quantities. This means that in learning per-trajectory constraints, we face more unidentifiability since there are more equivalent parameterizations (including the ones that are not decomposable) that produce the same set of optimal constrained policies. Further, if we had a per-trajectory constraint function c(\tau), we would have a much larger input space for the constraint function (which can now accept trajectories rather than a state-action pair). Additionally, with the same number of expert trajectories, we would also need considerably more datapoints for learning in order to prevent overfitting (since each trajectory is a datapoint, and thus we have fewer datapoints available compared to the case when we had state-action pairs). The constraint function would also need much more representational power to be able to process trajectories. And, with more representational power, searching for the right constraint function model will be more difficult.
>
> —----
>
> **References**
>
> [1] Review of Safe Reinforcement Learning: Methods, Theory and Applications, Gu et al. (2022)
>
> [2] Inverse Constrained Reinforcement Learning, Malik et al. (2021)

---

> > ### Author Response · Authors · 2022-11-16
> > **Response 2**
> >
> > > **I didn't understand the CartPole figures in Fig. 10. What do the "a=0" and "a=1" lines mean? What are the axes?**
> >
> > We have added an explanation for the CartPole figures in the relevant caption (Figure 12 now). Additionally, we have also labeled all the figures and axes and provided explanations, wherever possible. More specifically, the a=0 and a=1 lines represent the line for actions "go left" and "go right". For CartPole (MR), the pole must stay in x>=0.2 and for CartPole (Mid), the pole is not allowed to go right (action=1) for x>=1 and not allowed to go left (action=0) for x<=-1. The figures indicate the true constraint function. Here, the X-axis is pole position, and Y-axis is the constraint function value.
> >
> > > **How is the state space defined in the HighD environment? How does the agent observe the cars in front of it? How are the expert data collected?**
> > > **Detailed descriptions of the expert demonstrations used for their experiments are lacking.**
> >
> > For the HighD environment, the state space has 14 features. These consist of 7 ego features (x, y, speed, acceleration, rate of change of steering angle, steering angle and the heading angle) and 7 relevant predicates, propositions, and processed features (distance to the vehicle in front, processed distance, whether the ego is stopped, whether the ego has reached goal state, whether time limit has been exceeded, whether the ego is within road boundaries, and if ego has collided with another vehicle). We have added this information in Appendix C.1.
> >
> > Additionally, we have now added the information about expert dataset generation for all environments in Appendix C.9.
> >
> > > **It would be better to include the experimental details (in Appendix C.1) in the main paper.**
> >
> > Due to space requirements, it is difficult to include the experimental details in the main paper. However, we are ready to accommodate any specific restructuring changes that are suggested.

---

### Official Review · Reviewer_3D7X · 2022-10-23

**Confidence:** 4
**Clarity, Quality, Novelty And Reproducibility:** good
**Correctness:** 3
**Technical Novelty And Significance:** 3
**Empirical Novelty And Significance:** 3
**Recommendation:** 6

**Strength And Weaknesses:**

Strengths:
The problem of learning constraints is an important one and a solution to this problem can be impactful
The theoretical framework is helpful in gaining insight of the algorithm as well as providing theoretical justifications for the proposed algorithm
The proposed method shows decent performance on the selected tasks

Weaknesses:
The requirement of known reward function and single constraint is understandable but could limit the applicability of the algorithm to real-world problems.
The tasks used in the work are relatively simple.
The proposed computational algorithm seems quite complex with various tricks, such as the reweighting scheme, the two-stage constrained optimization with different weights, etc. This makes it potentially difficult to be applied to a wider variety of tasks, especially ones that are notably more complex.


**Summary Of The Paper:**

The paper proposed an inverse RL algorithm for recovering soft constraints from expert trajectories. The algorithm takes a nominal reward function, and expert trajectories as input, and outputs a recovered constraint function that best matches the behavior of the expert trajectories. The proposed algorithm interleaves between two steps: 1) solving a constrained RL problem with the latest constraint function, and 2) adjusting the constraint function to maximize the constraint value among all the previously obtained policies, subject to that expert trajectories satisfying the constraint. A theorem is provided to prove the convergence to the expert policy and a practical algorithm is provided that shows empirically successful results on three constraint learning tasks. The proposed method shows better performance in terms of constraint satisfaction as well as similarity to expert trajectories than baseline methods on the selected tasks.

**Summary Of The Review:**

My high-level concerns can be seen in the weaknesses section. One more detailed comment: the baseline method ICRL was applied to several more difficult continuous control tasks. It would more convincing if the proposed method was compared on similar tasks with the baselines.

Also, it is mentioned that the proposed method requires more training time than the baseline methods but I didn't find any numbers (I might have missed it). Could you provide some more details about how much slower it is?

In general the proposed method seems reasonable and attempts to solve an important problem. If the concerns above could be addressed it can be a good contribution to the ML community.


==============
post rebuttal: I have updated the score from 5 to 6 after reading other reviews and the authors' rebuttal.

---

> ### Author Response · Authors · 2022-11-16
> **Response 1**
>
> > **The requirement of known reward function and single constraint is understandable but could limit the applicability of the algorithm to real-world problems.**
>
> There are several reasons for using a known reward function and learning a single constraint function.
>
> First, this is a simpler setting compared to the setting when the reward function is unknown and when multiple constraint functions need to be learned. The latter setting has considerably more unidentifiability since there exist many more combinations of reward and constraint functions that achieve the given optimal policy (as mentioned in Section 6). In fact, at the time of our original submission, there was no existing literature (to the best of our knowledge) that handled this more complex setting, which can be attributed to the difficulty of this problem. More recently however, [6] has tried to tackle the problem of learning both a reward and per-agent constraint function in a multi-agent setting, assuming a linear parameterization of reward and constraint function. Since this is not yet scalable to complex setups and complex reward-constraint combinations, we believe that learning both arbitrary reward and (potentially multiple) constraints is still an open problem, especially in the single-agent setting.
>
> Second, as reviewer EuC1 noted, the setting of a single constraint function is not really limiting if the constraint can be arbitrarily complex (since it is a neural network). It may be possible to approximate arbitrarily closely multiple constraints by a single constraint with a more complex neural network.
>
> Finally, it is common to have knowledge of a nominal reward function in various settings. The behaviour not specified in the reward function may be specified better with a constraint function (Section 1). In particular, this work is indeed motivated by a potential real world application, i.e. application to autonomous driving, where the reward is typically known, but the constraint is not. In autonomous driving tasks, the reward can usually be defined to indicate some notion of progress, that is, the reward can be designed to ensure that we reach the destination in the minimum possible time (a small negative reward value for every timestep would be able to achieve this objective). However, the constraints which typically represent the bounds on quantities like gap to other vehicles, maximum possible velocity, acceleration or steering angle change are not trivially known in practice. Our method can find such bounds as we demonstrate with our HighD and ExiD experiments.
>
> > **The tasks used in the work are relatively simple.**
>
> > **One more detailed comment: the baseline method ICRL was applied to several more difficult continuous control tasks. It would more convincing if the proposed method was compared on similar tasks with the baselines.**
>
> As requested, we have conducted experiments with Ant and HalfCheetah environments, similar to ICRL paper [5], using a more challenging constraint compared to the ICRL paper. Additionally, we have also conducted experiments with the ExiD dataset (based on ~1000 trajectories of lane change data from 5 scenarios) by finding a constraint for lane change in highway driving. The corresponding experimental details are available in Appendix C, and the constraint/accrual plots are available in Appendix D. The discussion is available in Appendix B.6 and B.8. We hope these experiments can address the stated concerns.
>
> —----
>
> **References**
>
> [5] Inverse Constrained Reinforcement Learning, Malik et al. (2021)
>
> [6] Distributed Inverse Constrained Reinforcement Learning for Multi-agent Systems, Liu and Zhu (2022), https://openreview.net/forum?id=2Tv54LpM9cK

---

> > ### Author Response · Authors · 2022-11-16
> > **Response 2**
> >
> >
> > > **The proposed computational algorithm seems quite complex with various tricks, such as the reweighting scheme, the two-stage constrained optimization with different weights, etc. This makes it potentially difficult to be applied to a wider variety of tasks, especially ones that are notably more complex.**
> >
> > We would first like to point out that our design choices have been justified through ablation studies (see Section 5.2). More specifically, the reweighting scheme proposed in Section 4.3 does have an empirical advantage, which we have justified in Appendix B.2. For simplicity, the reweighting can be suspended (using uniform weights for the policy mixture) and the algorithm will still converge, but it may take more iterations to converge.
> >
> > The two-stage optimization is inspired from inverse RL (IRL) [1], and it is common in many IRL methods [2,3,4]. Our formulation in Equation (4) may be complex to optimize, which is why we propose the practically equivalent formulation in Equation (5), which we use in Algorithm 3.
> >
> > We have since conducted more experiments to address the concern regarding applicability to complex environments (Appendix B.6, B.8).
> >
> > > **Also, it is mentioned that the proposed method requires more training time than the baseline methods but I didn't find any numbers (I might have missed it). Could you provide some more details about how much slower it is?**
> >
> > We thank the reviewer for pointing this out, as this data wasn't already provided. As requested, we have added training time statistics in Table 12. The training time is further discussed in Appendix C.8. The proposed method should require more training time *in principle* as discussed in Appendix C.8, but due to implementation differences between our code and baselines, we empirically find that the average training time of ICL is roughly comparable to the baselines.
> >
> > —----
> >
> > **References**
> >
> > [1] A Survey of Inverse Reinforcement Learning: Challenges, Methods and Progress, Arora & Doshi (2021)
> >
> > [2] Maximum entropy inverse reinforcement learning, Ziebart et al. (2008)
> >
> > [3] Learning robust rewards with adversarial inverse reinforcement learning, Fu et al. (2017)
> >
> > [4] Guided cost learning: Deep inverse optimal control via policy optimization, Finn et al. (2016)

---

> > > ### Comment · Reviewer_3D7X · 2022-11-21
> > > **thanks for the rebuttal**
> > >
> > > The authors' response addressed most of my main concerns. The additional experiment as well as the timing statistics are greatly appreciated. I will update my score to 6 from 5.

---

### Official Review · Reviewer_8Rs7 · 2022-10-23

**Confidence:** 3
**Correctness:** 3
**Technical Novelty And Significance:** 3
**Empirical Novelty And Significance:** 2
**Recommendation:** 5

**Clarity, Quality, Novelty And Reproducibility:**

The overall clarity, quality, and novelty are good. Can't comment on the reproducibility.

**Strength And Weaknesses:**

Strength:
The paper provides a novel framework with a theoretical formulation and a practical algorithm with reasonable relaxations. It also provides empirical results to show the improvements.

Weaknesses:
1. For the definition of 'soft' constraint, as described in future work, I also feel that the constraint through expectation is not as useful as constraint through probability for stochastic environments. Can the author give more explanation of where the current setting could be applied in real life?

2. It would be great if the author could explain more about how this 'soft' constraint is guiding the algorithm design.

2. The experiment results seems not strong enough to me. There is only one real-work environment and the setting is also relatively simple. It would be good to provide more complex experiment setups (e.g. robotics manipulations).

**Summary Of The Paper:**

This paper proposes a novel framework to learn soft constraints for ICL. The paper gives a theoretical formulation with proof of convergence, and then provides a practical algorithm with relaxation on the theoretical formulation and several techniques. Experiments on synthetic and real-world environments show improvements of the proposed method compared to baselines. Some ablation studies are provided to verify the design choices of the algorithm.

**Summary Of The Review:**

I think the paper provides an interesting framework with a practical algorithm. Each part of the paper is 'good' but not 'great' with the weaknesses mentioned above.

---

> ### Author Response · Authors · 2022-11-16
> **Response 1**
>
> > **For the definition of 'soft' constraint, as described in future work, I also feel that the constraint through expectation is not as useful as constraint through probability for stochastic environments. Can the author give more explanation of where the current setting could be applied in real life?**
>
> We first elaborate on some concrete applications of the current setting:
>
> (a) **Inventory planning**: For products in an inventory, the goal could be to minimize holding and backlog cost for the products while satisfying the constraint that expected resource consumption of the products is within a budget ([1], Section 6).
>
> (b) **Queue scheduling in network routing**: Given pools of servers with different skillsets and customers who wish to use the various servers, the customers typically wait in queue until they are allocated a server. The objective is to find a scheduling policy such that the cost of holding customers in the queue and routing costs are minimized subject to capacity constraints, which leads to an expected constraint formulation ([1], Section 7).
>
> (c) **Robotics**: Many robotics settings have expected safety constraints [2,3,4,5,6,9]
>
> (d) **Autonomous driving**: In motion planning, the goal could be to reach a destination in shortest time, while having an expected constraint on the number of obstacles encountered ([8], Section 5 and [9], Section 5). More practically, we could also bound the expected amount of time that the agent spends really close to an obstacle.
>
> (e) **Video compression**: For online streaming services, it could be useful to optimize the video quality subject to expected constraints on the bitrate [13].
>
> We further refer the reviewer to [10] for an extensive survey on works that use this setting. In fact, we would like to mention that the two real world highway driving experiments (HighD and ExiD) used in this work were actually requested by our industrial partner. These experiments find constraints for autonomous driving settings like maintaining a gap to the vehicle in the front and lane change, which are not trivially known in practice. These constraints will be used in their motion planning software stack.
>
> Finally, we would like to note that the setting of expected constraints is well established in constrained RL [11]. Expected constraints are a more general formulation, and can implicitly express certain probabilistic and hard constraints. More concretely, if the per-step constraint function c(s,a) is an indicator function over some state-action set, then bounding the expectation is equivalent to bounding the probability of entering this set, which is equivalent to a probabilistic constraint. And, with β=0 and any non-negative c(s,a), the expected constraint is equivalent to a hard constraint since the agent must avoid all state-action pairs with positive constraint value at all times to satisfy the expected constraint with β=0 (see [12], Section 2.1.3.3).
>
> —---------
>
> **References**
>
> [1] A Primal-Dual Approach to Constrained Markov Decision Processes, Chen et al. (2021)
>
> [2] Accelerated Primal-Dual Policy Optimization for Safe Reinforcement Learning, Liang et al. (2018)
>
> [3] Constrained Markov Decision Processes via Backward Value Functions, Satija et al. (2020)
>
> [4] Reward Constrained Policy Optimization, Tessler et al. (2018)
>
> [5] Constrained Policy Optimization, Achiam et al. (2017)
>
> [6] Reinforcement Learning Control of Constrained Dynamic Systems with Uniformly Ultimate Boundedness Stability Guarantee, Han et al. (2020)
>
> [7] Projection-Based Constrained Policy Optimization, Yang et al. (2020)
>
> [8] Lyapunov-based Approach to Safe Reinforcement Learning, Chow et al. (2018)
>
> [9] Lyapunov-based Safe Policy Optimization for Continuous Control, Chow et al. (2019)
>
> [10]  Review of Safe Reinforcement Learning: Methods, Theory and Applications, Gu et al. (2022)
>
> [11] Constrained Markov Decision Processes, Altman et al. (2004)
>
> [12] Safe Learning in Robotics: From Learning-Based Control to Safe Reinforcement Learning, Brunke et al. (2022)
>
> [13] MuZero with Self-competition for Rate Control in VP9 Video Compression, Mandhane et al. (2022)

---

> > ### Author Response · Authors · 2022-11-16
> > **Response 2**
> >
> > > **It would be great if the author could explain more about how this 'soft' constraint is guiding the algorithm design.**
> >
> > Our algorithm design (Algorithms 1-3) is motivated by Equations (3), (4) and (5), all of which use the expected (soft) constraints instead of the hard constraints prevalent in constraint learning literature. More specifically, if we had a hard constraint, we would need to ensure that the constraint holds for each trajectory, and not on average across the trajectories.
> >
> > > **The experiment results seems not strong enough to me. There is only one real-work environment and the setting is also relatively simple. It would be good to provide more complex experiment setups (e.g. robotics manipulations).**
> >
> > As requested, we have conducted experiments with Ant and HalfCheetah environments, similar to ICRL paper [14]. Additionally, we have also conducted experiments with the ExiD dataset by finding a constraint for lane change in highway driving. The corresponding experimental details are available in Appendix C, and the constraint/accrual plots are available in Appendix D. The discussion is available in Appendix B.6 and B.8. With these environments, we now have evaluated our method on 8 environments, 4 of which are robotics related or autonomous driving related. We hope these experiments are sufficient to address the stated concerns.
> >
> > —---------
> >
> > **References**
> >
> > [14] Inverse Constrained Reinforcement Learning, Malik et al. (2021)

---

### Official Review · Reviewer_EuC1 · 2022-10-24

**Confidence:** 3
**Correctness:** 3
**Technical Novelty And Significance:** 4
**Empirical Novelty And Significance:** Not applicable
**Recommendation:** 8

**Clarity, Quality, Novelty And Reproducibility:**

The proposed method appears to be novel, and very different from the few previous methods known for solving this class of problems. The paper is written very well and is largely self-contained. I would suggest, though, to explain in plain text the logic behind the constraints in the experimental environments. For example, what is the meaning of the constraints on the cart pole systems in Fig. 10, and what is the optimal policy under them? It is difficult to figure it out just by looking at the plots of the constraint function. (And, it does not help that the coordinate axes do not have labels or dimensions.)

**Strength And Weaknesses:**

The paper proposes a principled way of addressing the problem, borrowing from the computational principle and machinery of inverse reinforcement learning (IRL) methods, and the empirical evaluation appears to be convincing. The limitation that there is only one constraint is probably not significant, as this constraint could potentially be an arbitrarily complex function on state/action pairs. (Unlike many IRL algorithms, where the learned reward function is linear in the vector of parameters.) It appears that the chosen parametric form of the constraint function is a neural net with two hidden layers of 64 units each, ReLU activations, and final output with a sigmoid activation function (Table 5). Why this form, and is it really sufficient to represent arbitrary functions? What about the constraints in the chosen experimental environments, how well does it represent its constraints? Maybe learning the constraints directly from the known functions by means of supervised learning could answer this question, and also effectively provide a lower bound on the MSE that this method can provide?

Another thing I was not clear about is the effect of the initial form of the constraint (that is, c in Equation 3 on the very first iteration of the algorithm). Was it completely random in the computational experiments, and does its value matter? In other words, does the algorithm always converge to a global minimum, or there are local minima?

Some minor typos:
P.1: "are therefore are" -> "therefore are"
P.3: "in an other line of work" -> "in another line of work"



**Summary Of The Paper:**

The paper proposes a method for learning soft constraints in sequential decision making problems that can be modeled as MDPs, under the assumption that the entire MDP model is known (both transition probabilities and the reward function), but a constraint on the state-action space exists that is known only to a decision maker who can provide optimal decisions that respect this constraint, and the objective is to recover the constraint from the observed decisions of the expert. Empirical evaluation on several domains demonstrates that the proposed method is far superior to estimating the constraint than two existing methods.

**Summary Of The Review:**

The paper proposes a novel method for recovering soft constraints in sequential decision problems, where the MDP model is known, and expert demonstrations are available under the unknown constraint. The empirical evaluation demonstrates that the method has much lower MSE in recovering the unknown constraint that two existing baseline methods.

---

> ### Author Response · Authors · 2022-11-16
> **Response 1**
>
> > **It appears that the chosen parametric form of the constraint function is a neural net with two hidden layers of 64 units each, ReLU activations, and final output with a sigmoid activation function (Table 5). Why this form, and is it really sufficient to represent arbitrary functions? What about the constraints in the chosen experimental environments, how well does it represent its constraints?**
>
> For our experimental settings, the chosen constraint function architecture is able to represent the constraints well, as supported by our low CMSE scores (Tables 1, 5) and recovered constraint plots (Appendix D). Our architectural choice is not new, as this specific architecture is also assumed by the baselines (GAIL-Constraint and ICRL) in the ICRL paper ([4], Section 3.2).
>
> We note that the number of layers, hidden neurons and intermediate activations may affect the class of constraint functions that can be learned with this architecture. The architecture for the constraint function can be changed to anything desired by the designer in order to approximate an actual constraint arbitrarily closely. The two-layer architecture mentioned in the paper is not a requirement but simply an example that was sufficient for our experiments.
>
> Increasing the flexibility of the constraint space that can be represented by an architecture will also increase the unidentifiability of the problem. More precisely, there can be multiple constraint functions (and more parameterizations) that induce the same given optimal policy [2], similar to IRL where multiple reward functions may induce the same given optimal policy [1]. In the case of rewards, there are two types of unidentifiability: representational and experimental, as discussed in [3]. Representational unidentifiability refers to the fact that for any reward R, a reward aR+c (scaled and shifted) obtains the same optimal policies (so it is hard to distinguish between R and aR+c), while experimental unidentifiability occurs when there are multiple structurally different rewards that produce the same given demonstrations, and it is impossible to discern the correct reward with a single experiment. The authors of [3] resolve representational unidentifiability by imposing structure on the reward, that is, the reward can only assume a canonical representation (being in between 0 and 1). This doesn't change the set of optimal policies. Similarly, in the case of constraint functions, having arbitrary constraint values is akin to having representational unidentifiability. Imposing a structure (NN with sigmoid output) on the constraint function is then equivalent to learning a canonical constraint function. By fixing this structure, the constrained behaviour is determined primarily by β rather than the arbitrary constraint value magnitude. Proving formally that having a canonical representation yields the same set of optimal constrained policies would be an interesting future direction.
>
> Finally, another motivation for choosing this structure for the constraint function arises from the fact that outputs in [0, 1] may be interpreted as safeness values in many applications (as mentioned in Section 5). A constraint value of 0 can be viewed as a relatively safe state-action pair, while a constraint value of 1 can be viewed as an unsafe state-action pair. In comparison, arbitrary constraint values are hard to interpret.
>
>
> —-------
>
> **References**
>
> [1] Algorithms for Inverse Reinforcement Learning, Ng and Russell (2000)
>
> [2] Learning Parametric Constraints in High Dimensions from Demonstrations, Chou et al. (2020)
>
> [3] Towards Resolving Unidentifiability in Inverse Reinforcement Learning, Amin and Singh (2016)
>
> [4] Inverse Constrained Reinforcement Learning, Malik et al. (2021)

---

> > ### Author Response · Authors · 2022-11-16
> > **Response 2**
> >
> > > **Maybe learning the constraints directly from the known functions by means of supervised learning could answer this question, and also effectively provide a lower bound on the MSE that this method can provide?**
> >
> > We agree that given a particular constraint function architecture, training to minimize error w.r.t. a known true constraint function can be used to produce a lower bound on the CMSE. Intuitively, an architecture with more representational power will lead to a lower value of this lower bound.
> >
> > > **Another thing I was not clear about is the effect of the initial form of the constraint (that is, c in Equation 3 on the very first iteration of the algorithm). Was it completely random in the computational experiments, and does its value matter? In other words, does the algorithm always converge to a global minimum, or there are local minima?**
> >
> > The constraint function is randomly initialized for every ICL run, and its value may affect the time needed for convergence and the quality of the recovered constraint. Since this is a nonconvex problem, the algorithm may converge to local minima. Note that for most of deep learning, since the optimization problems are typically non-convex there is no guarantee of convergence to global optima and our approach is in line with this state of affairs.
> >
> > > **Some minor typos: P.1: "are therefore are" -> "therefore are" P.3: "in an other line of work" -> "in another line of work"**
> >
> > We thank the reviewer for pointing out these typos. We have incorporated the suggestions in the updated paper.
> >
> > > **I would suggest, though, to explain in plain text the logic behind the constraints in the experimental environments. For example, what is the meaning of the constraints on the cart pole systems in Fig. 10, and what is the optimal policy under them? It is difficult to figure it out just by looking at the plots of the constraint function. (And, it does not help that the coordinate axes do not have labels or dimensions.)**
> >
> > We have added an explanation for the CartPole constraints (and the optimal policy) in the relevant figure (Figure 12 now). We have also labeled all the figures and axes and provided explanations, wherever possible. To clarify for the CartPole systems: for CartPole (MR), the pole must stay in x>=0.2 and for CartPole (Mid), the pole is not allowed to go right (action=1) for x>=1 and not allowed to go left (action=0) for x<=-1.

---

> > > ### Comment · Reviewer_EuC1 · 2022-11-17
> > > **Response to the authors' response**
> > >
> > > Thank you for clarifying why you chose this networks architecture for representing the constraint, and for improving the interpretability of the figures related to the empirical validation. Regarding learning the constraint in a supervised learning manner by sampling training examples from the known equation, my suggestion was to actually do the experiment and report the results in the paper, in order to test the representational power of this network and also establish a lower bound of sorts for the error of the proposed main algorithm.

---

> > > > ### Author Response · Authors · 2022-11-18
> > > > **Response**
> > > >
> > > > Thank you for clarifying your suggestions. As requested, we have added experiments to compute the lower bounds in Appendix C.10. Our results are reported in Tables 13 and 14. As can be seen from the tables, having too few hidden nodes or layers is not sufficient to arbitrarily approximate the true constraint function. Our architectural choice is able to achieve a lower bound of 0.00 ± 0.00, accurate up to two decimals. So it is a valid choice and the architecture has sufficient representational capacity.

---

> > > > > ### Comment · Reviewer_EuC1 · 2022-11-18
> > > > > **Added experimental results**
> > > > >
> > > > > Thank you for adding this well designed experiment. It clearly demonstrates that the chosen neural network architecture is sufficiently complex for representing accurately the constraints in all  test environments, and also indicates that simpler architectures, with fewer hidden layers and/or fewer units in them, might not be sufficient for some problems.

---

### Author Response · Authors · 2022-11-16
**General response**

We thank the reviewers for their constructive and positive feedback. We have changed the paper to accommodate the concerns raised by the reviewers. Most of these changes are in the Appendix, but some changes are also in the main paper. *All changes are highlighted in blue colour*. Here is a summary of these changes:

(a) **Experiments with Mujoco environments and ExiD lane change environment**: As requested, we have conducted experiments with Ant and HalfCheetah environments, similar to ICRL paper [1]. We note that there are two other environments in the ICRL paper, but they use implicit constraints, hence they cannot be used with our algorithm. To compensate for this, we make the environments more challenging by adjusting the constraint. Additionally, we also conduct experiments with the ExiD dataset by finding a constraint for lane change in highway driving. This environment is constructed based on ~1000 trajectories spanning 5 different scenarios, which is more realistic compared to our existing experiment with HighD dataset (based on ~100 trajectories). The corresponding experimental details are available in Appendix C, and the constraint/accrual plots are available in Appendix D. The discussion is available in Appendix B.6 and B.8. To conduct these additional experiments, we integrated OpenAI's (highly efficient) safety-agents implementation [2] within our ICL algorithm. The hyperparameter section (Appendix C.5-C.7) has also been updated to include the details of this implementation.

(b) **Training time statistics**: As requested by reviewer 3D7X, we have added Appendix C.8 and Table 12 to report the training times for our experiments. We find that our implementations of ICL require an overall average runtime that is comparable to the baselines, which establishes their efficiency and viability for practical use.

(c) **Restructuring and minor additions**: To incorporate the information about the new environments and stay within the page limit (for the main paper), we moved the discussion for HighD driving environment to Appendix B.7. We have also labeled all the figures and added explanations to captions in some places to improve clarity. Additionally, we have also clarified how the expert trajectories are generated (requested by reviewer 5hKt) in Appendix C.9. We have also added further explanation regarding the use of other constrained RL implementations within the ICL algorithm in Section 4.2. We are open to further suggestions regarding restructuring other parts of the paper.

We would also like to briefly reiterate our contributions. Our paper proposes Inverse Constraint Learning (ICL), a method to learn expected/soft constraints from expert data. To the best of our knowledge, this is the first work to propose a method for learning soft constraints. These constraints are satisfied in expectation, which means they can be occasionally violated in order to achieve the objective at hand. This is a more general type of constraint than hard and probabilistic constraints, and learning soft constraints has many real world applications in autonomous driving, robotics, network routing, video compression, etc. We demonstrate our approach with (now) 8 environments of varying difficulty, where 4 of these environments are related to robotics and real world tasks in autonomous driving. We hope that performance in these settings is sufficiently indicative of the viability of this algorithm.


—------

**References**:

[1] Inverse Constrained Reinforcement Learning, Malik et al. (2021)

[2] Benchmarking Safe Exploration in Deep Reinforcement Learning, Ray et al. (2019)

---

### Decision · Program_Chairs · 2023-01-20

**Decision:**

Accept: notable-top-25%

**Justification For Why Not Higher Score:**

The paper doesn't seem to be sufficiently ground-breaking to justify an oral presentation.

**Justification For Why Not Lower Score:**

 I think the setting is a good one and the method seems smart and the experiments good.  The reviewers were on balance positive.

Poster would also be fine, but I feel it should be accepted.

**Metareview: Summary, Strengths And Weaknesses:**

This paper addresses a variant of the inverse reinforcement-learning problem in the context of constrained MDPs.  In this setting, there is a generic known reward function but the interesting problem is to infer the constraints that the demonstrator is operating under.

I believe this is an important setting which may be more useful for robotics than "standard" IRL, because constraints are both more generic (keeping my waterglass upright can be done with all kinds of different robots and strategies) and composable (if I can infer two constraints from two different sets of demonstrations, I can independently solve for a strategy that tries to satisfy both of them.)  The paper is well written and reviewers find it to be novel.  The authors provided a very nice list of applications in which this setting is appropriate.

Initially there were weaknesses with respect to the experimental validation, but the reviewers suggested additional experiments and the authors carried them out effectively.  Some additional concerns about writing and formulation were largely addressed in rebuttals.




**Note From Pc:**

if the above contains the word "oral" or "spotlight" please see: "oral" presentation means -> notable-top-5% and "spotlight" means -> notable-top-25%. As stated in our emails, we are disassociating presentation type from AC recommendations